# Cholinergic modulation shifts the response of CA1 pyramidal cells to depolarizing ramps via TRPM4 channels with potential implications for place field firing

Crescent L Combe[1], Carol M Upchurch[2], Carmen C Canavier[2]*, Sonia Gasparini[1,2]*

[1]Neuroscience Center of Excellence, Louisiana State University Health Sciences Center, New Orleans, United States; [2]Department of Cell Biology and Anatomy, School of Medicine, Louisiana State University Health Sciences Center, New Orleans, United States

*For correspondence:
ccanav@lsuhsc.edu (CCC);
sgaspa1@lsuhsc.edu (SG)

Competing interest: The authors declare that no competing interests exist.

**Abstract** A synergistic combination of in vitro electrophysiology and multicompartmental modeling of rat CA1 pyramidal neurons identified TRPM4 channels as major drivers of cholinergic modulation of the firing rate during a triangular current ramp, which emulates the bump in synaptic input received while traversing the place field. In control, fewer spikes at lower frequencies are elicited on the down-ramp compared to the up-ramp due to long-term inactivation of the $Na_V$ channel. The cholinergic agonist carbachol (CCh) removes or even reverses this spike rate adaptation, causing more spikes to be elicited on the down-ramp than the up-ramp. CCh application during Schaffer collateral stimulation designed to simulate a ramp produces similar shifts in the center of mass of firing to later in the ramp. The non-specific TRP antagonist flufenamic acid and the TRPM4-specific blockers CBA and 9-phenanthrol, but not the TRPC-specific antagonist SKF96365, reverse the effect of CCh; this implicates the $Ca^{2+}$-activated nonspecific cation current, $I_{CAN}$, carried by TRPM4 channels. The cholinergic shift of the center of mass of firing is prevented by strong intracellular $Ca^{2+}$ buffering but not by antagonists for $IP_3$ and ryanodine receptors, ruling out a role for known mechanisms of release from intracellular $Ca^{2+}$ stores. Pharmacology combined with modeling suggest that $[Ca^{2+}]$ in a nanodomain near the TRPM4 channel is elevated through an unknown source that requires both muscarinic receptor activation and depolarization-induced $Ca^{2+}$ influx during the ramp. Activation of the regenerative inward TRPM4 current in the model qualitatively replicates and provides putative underlying mechanisms for the experimental observations.

## Editor's evaluation

This manuscript by Combe et al. presents the role of cholinergic modulation in the spike rate adaptation in pyramidal place cells. Using combined electrophysiology, pharmacological, and multi-compartment computational modeling, the authors identify the downstream pathway (e.g. activation of TRPM4 channel) that shapes the firing pattern under the triangular-shaped ramps. The study demonstrates solid evidence, and these rigorous findings are important for bridging pyramidal neurons' molecular/channel properties to behavior-level implications (place field firing).

## Introduction

Release of the neuromodulator acetylcholine is thought to be correlated with certain brain states or functions (*Pepeu and Giovannini, 2004*). In the hippocampal formation, elevated acetylcholine is associated with exploration (*Bianchi et al., 2003*), particular memory processes (*Micheau and Marighetto, 2011*), REM sleep (*Hutchison and Rathore, 2015*), and response to novelty (*Acquas et al., 1996*; *Thiel et al., 1998*; *Giovannini et al., 2001*).

The major cholinergic input to the hippocampus is from the medial septum and diagonal band of Broca, which innervate all hippocampal subfields (*Dutar et al., 1995*). As for the degree to which acetylcholine release is spatially targeted, this is still under debate (*Disney and Higley, 2020*; *Sarter and Lustig, 2020*); however, it has been shown that CA1 receives cholinergic afferents in all layers, although at different densities (*Nyakas et al., 1987*; *Aznavour et al., 2002*). Pyramidal neurons express G-protein-coupled metabotropic muscarinic acetylcholine receptors (mAChRs; *Levey et al., 1995*), more so than they do ionotropic (nicotinic) receptors, which are primarily on interneurons in this region (*Dani and Bertrand, 2007*). $M_1$ mAChRs, in particular, are found on both dendritic shafts and spines of CA1 pyramidal neurons (*Yamasaki et al., 2010*).

Elevated acetylcholine in the hippocampal formation is thought to favor memory encoding, and modulates theta oscillations (*Hasselmo, 2006*), which in turn modulate place cell firing (*Easton et al., 2012*). There is evidence that cholinergic activity regulates long-term synaptic plasticity (*Huerta and Lisman, 1995*; *Palacios-Filardo and Mellor, 2019*; *Fernández de Sevilla et al., 2021*). Cholinergic activation also influences intrinsic properties of neurons, which result in membrane potential depolarization (*Brown and Adams, 1980*), increase in firing rate (*Benardo and Prince, 1982*), and modulation of input/output relationships, for example switching from burst to regular firing (*Azouz et al., 1994*).

In addition to suppressing several potassium currents (*Brown and Adams, 1980*; *Hoffman and Johnston, 1998*; *Buchanan et al., 2010*; *Giessel and Sabatini, 2010*), cholinergic stimulation has been linked to the activation of a calcium-activated, non-specific cation current, $I_{CAN}$ (*Colino and Halliwell, 1993*; *Egorov et al., 2002*; *Yoshida et al., 2012*). $I_{CAN}$ is attributed to current flowing through channels of the transient receptor potential (TRP) family (*Guinamard et al., 2010*; *Reboreda et al., 2011*). The canonical TRP channel, TRPC, has been implicated in cholinergic-mediated persistent firing (*Zhang et al., 2011*; *Arboit et al., 2020*), but see *Egorov et al., 2019*. There is indirect evidence, however, of involvement of TRPM4 channels in the cholinergic activation of $I_{CAN}$. TRPM4 channel activation downstream of group I metabotropic glutamate receptors in pre-Botzinger cells (*Mironov, 2008*), implicates the $G_{q/11}$ pathway shared by $mGluR_{1/5}$ and mAChR $M_{1/3/5}$. Although TRPM4 channels are known to be expressed in the soma and apical dendrites of CA1 pyramidal dendrites (*Riquelme et al., 2021*), their known functions in these neurons are currently limited to a minor role in the after-depolarization following action potential trains (*Lei et al., 2014*) and involvement in NMDA-mediated LTP (*Menigoz et al., 2016*).

Previously, we found that when triangular shaped current ramps intended to emulate the depolarizing input received by a place cell during a place field traversal (*Epsztein et al., 2011*) were injected at the soma or dendrites of CA1 pyramidal neurons, they elicited fewer spikes on the down-ramp than on the up-ramp (*Upchurch et al., 2022*). In the current study, we investigated the effect of cholinergic modulation on this firing pattern in vitro (slice electrophysiology) and in silico (multi-compartmental NEURON model). We found that, at comparable maximum firing rates, cholinergic activation shifts the center of mass of firing to the latter half of the ramp, both for current injection ramps as well as for synaptically-driven depolarizations. Both CCh and sustained depolarization are required to produce the rightward shift in the center of mass of firing along the ramp. This shift is blocked by antagonists specific for TRPM4 channels or by intracellular $Ca^{2+}$ buffering, but not by $IP_3$ or ryanodine receptor antagonists. The TRPM4 channels have a micromolar half-activation concentration of $Ca^{2+}$ (*Nilius et al., 2004*), which implies that a restricted $Ca^{2+}$ nanodomain near the channel mouth is required to elicit substantial opening of TRPM4 channels. Thus, whatever the $Ca^{2+}$ source that activates the TRPM4 channels, it is likely co-localized with them.

# Results

## Asymmetric firing on symmetric ramps is modulated by cholinergic activation

We have previously used 2 and 10 s long triangular-shaped current ramps (*Upchurch et al., 2022*) to emulate the depolarizing input received by a place cell during a place field traversal (*Epsztein et al., 2011*); the amplitude of the ramps was adjusted for each cell to produce peak frequencies between 10 and 25 Hz, as observed in place cells in vivo (*Hargreaves et al., 2007*; *Resnik et al., 2012*; *Bittner et al., 2015*). Regardless of whether these ramps are injected at the soma or in the dendrites of CA1 pyramidal neurons, they elicit fewer spikes on the down-ramp than on the up-ramp, due to adaptation mediated by long-term sodium channel inactivation (*Fernandez and White, 2010*; *Venkatesan et al., 2014*; *Upchurch et al., 2022*). Place cell firing is modulated by experience such that the center of mass of the place field shifts in a direction opposite the direction of motion as the environment becomes more familiar (*Mehta et al., 2000*). This modulation is strongest when animals are first introduced to a novel environment (*Roth et al., 2012*), suggesting involvement of a novelty signal. Since elevated acetylcholine is thought to be such a signal (*Acquas et al., 1996*; *Giovannini et al., 2001*), we investigated the effect of cholinergic neuromodulation on this firing rate adaptation (*Figure 1*).

*Figure 1A1 and E1* show typical voltage responses to two-second-long symmetric ramps injected in the soma or dendrites (approximately 200 µm from soma) under control conditions. This type of response results in a positive value for the normalized difference between spikes fired on the up-ramp versus the down-ramp (we call this an adaptation index, see Materials and methods for definition). A positive index means that neurons fire more on the up-ramp than the down-ramp, whereas a negative value indicates the converse, and is therefore a good metric to determine shifts in the center of mass of the firing of CA1 pyramidal neurons in different conditions. In some trials, under control conditions we applied a baseline depolarization prior to the ramp, in order to capture the variability of the membrane potential observed in vivo (*Harvey et al., 2009*; *Epsztein et al., 2011*). Application of the cholinergic agonist carbachol (CCh, 2 µM) caused a depolarization of 2–6 mV. We compensated for this depolarization by injecting tonic hyperpolarizing current to reestablish the original membrane potential (see also *Losonczy et al., 2008*), as indicated by an offset from the 0 pA current level in the traces of the injected current ramps. The amplitude of background fluctuations in the resting membrane potential increased from a few tenths of a mV in control to 2–4 mV in CCh. Moreover, the threshold for action potential generation became more hyperpolarized. For all these reasons, we were not able to consistently vary the membrane potential using baseline depolarizations in the presence of CCh, because baseline depolarization alone frequently evoked spiking. As a result of the increased excitability in CCh, smaller current ramps were sufficient to produce the same peak frequencies as in control (note the difference in the injected ramp currents below the voltage trace in *Figure 1A1 vs. B1 and E1 vs. F1*). Under these conditions, CCh caused a shift of the center of mass of firing to later in the ramp (*Figure 1B1 and 1F1*), which resulted in a decreased value of the adaptation index which became negative for most cells, as shown in the scatter plot of group data (*Figure 1D and H*). The shift is evident in the plots of the frequency as a function of time (compare *Figure 1A2 vs. 1B2 and E2 vs. F2*). As in *Mainen and Sejnowski, 1995*, the raster plots of the spike times were characterized by variability in the timing of action potentials among different trials in the same neuron (*Figure 1A3, B3, E3 and F3*). However, the variability appears to be independent of the amount of current injected, as demonstrated by trials that are grouped as a function of the injected peak current amplitude. Moreover, the raster plots show how the center of mass of firing overall shifted from early in the ramp in control conditions to later in the ramp in the presence of carbachol. This shift was significant for short (2 s) ramps in the soma (control 0.33±0.02, CCh –0.25±0.03; t(20) = 16.741; p<0.0005; n=21), long (10 s) ramps in the soma (control 0.24±0.02, CCh –0.33±0.06; t(18) = 9.854; p<0.0005; n=19), as well as for short and long ramps in the dendrites (2 s ramps: control 0.49±0.04, CCh –0.45±0.07; t(18) = 12.529; p<0.0005; n=19; 10 s ramps: control 0.37±0.02, CCh –0.16±0.06; t(9) = 10.582; p<0.0005; n=10).

Plots of instantaneous frequency versus current injection are shown in *Figure 1C1, C2, G1 and G2*. These plots provide a way to assess firing rate adaptation or acceleration when the injected current is not a square pulse (*Venugopal et al., 2015*). In control (*Figure 1C1 and G1*), frequencies were higher on the up-ramp (filled circles) and decreased on the down-ramp (open circles) at similar values of injected current. This hysteretic clockwise motion corresponds to spike rate adaptation, in which

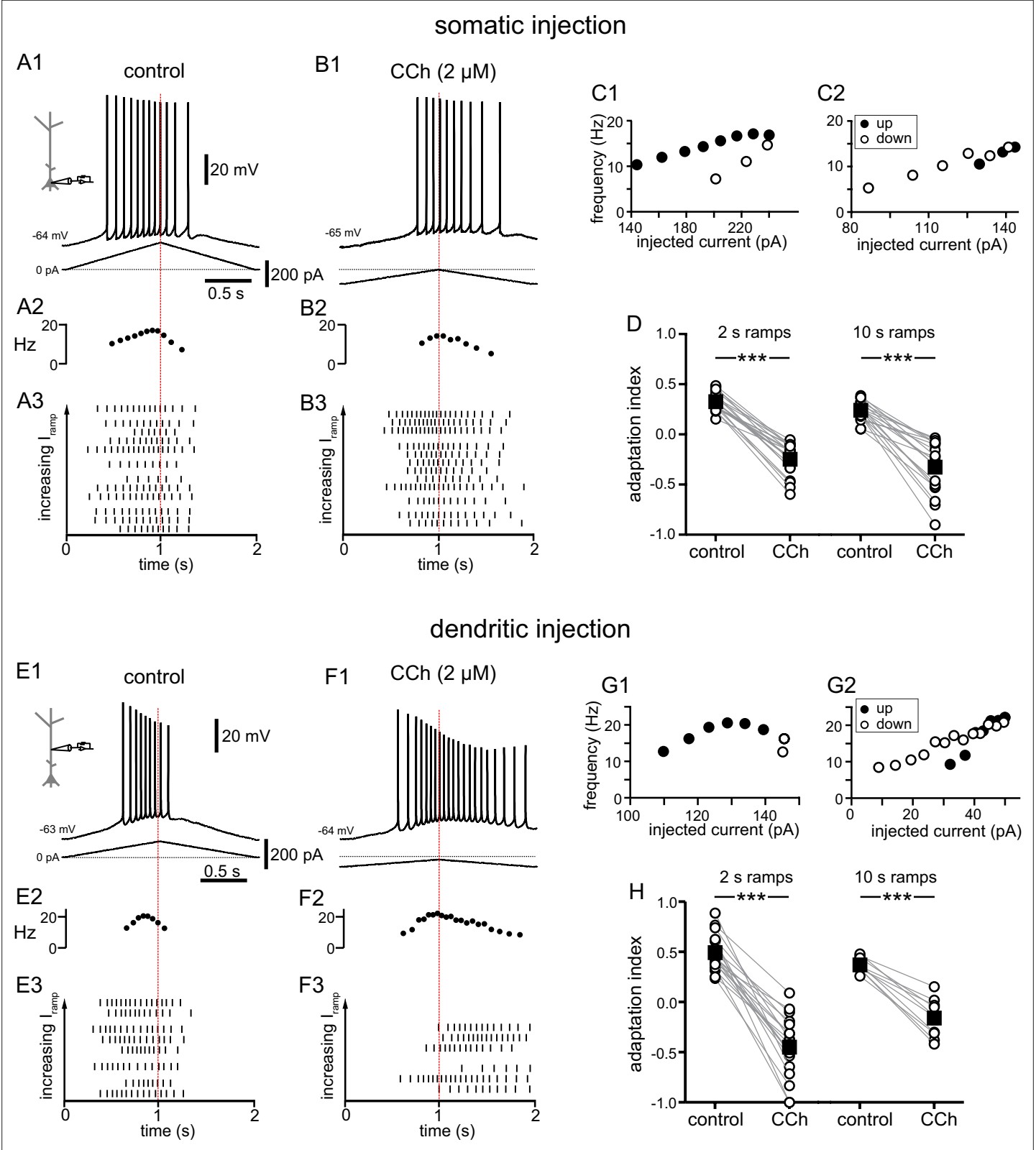

**Figure 1.** Carbachol shifts the center of mass of firing in response to depolarizing current ramps. (**A**) Somatic recording, control. (**A1**) Voltage trace for a two second triangular current ramp (represented below) injected in the soma. (**A2**) Instantaneous frequency for example in A1 as it changes along the ramp. (**A3**) Raster plot of different trials recorded from the same example neuron in A1. The trials are ordered by the amplitude of the peak injected current, from lowest (bottom) to highest (top); trials with the same amount of current injection are clustered together. (**B**) Somatic recording, carbachol.

*Figure 1 continued on next page*

*Figure 1 continued*

(**B1**, **B2**, and **B3**) Same as A1, A2, and A3, but during superfusion with 2 µM CCh. (**C1**) Instantaneous frequency for example in A1, as it changes with injected current. Filled circles represent firing on the up-ramp; open circles, firing on the down-ramp. (**C2**) Instantaneous frequency for example in B1. **A**, **B**, and **C** are recordings from the same cell. (**D**) Summary data of the adaptation index for somatic recordings, before and during CCh, for two second ramps (n=21) and ten second ramps (n=19). Open circles connected by gray lines represent indices for individual cells, before and during CCh. Black squares with error bars represent group averages ± SEM. In control, all adaptation indices are positive, indicating more action potentials on the up-ramp. In CCh, most adaptation indices are negative, indicating more action potentials on the down-ramp. (**E**) Dendritic recording, control. (**E1**) Voltage trace for a two second triangular current ramp (represented below) injected in the dendrite, approximately 200 µm from the soma. (**E2**) Instantaneous frequency for E1 as it changes along the ramp. (**E3**) Raster plot of different trials recorded from same example neuron in E1. (**F**) Dendritic recording, carbachol. (**F1**, **F2**, and **F3**) Same as E1, E2, and E3, but during superfusion with 2 µM CCh. (**G1**) Instantaneous frequency for example in E1, as it changes with injected current. (**G2**) Instantaneous frequency for example in F1. **E**, **F**, and **G** are from the same cell. (**H**) Summary data of the adaptation index for dendritic ramps, before and during CCh, for two second ramps (n=19) and ten second ramps (n=10). *** p<0.0005, paired t-test. Red dotted lines in the left panels mark the middle of the current injection ramp. Source data in "*Figure 1—source data 1*". See also *Figure 1—figure supplement 1* and *Figure 1—figure supplement 2*.

The online version of this article includes the following source data and figure supplement(s) for figure 1:

**Source data 1.** Carbachol shifts the center of mass of firing in response to depolarizing current ramps.

**Figure supplement 1.** Plots of instantaneous frequency versus current injection showing different effects of carbachol on firing rate adaptation.

**Figure supplement 2.** Effects of the membrane potential, injected current and input resistance on the adaptation index.

**Figure supplement 2—source data 1.** Carbachol shifts the center of mass of firing in response to depolarizing current ramps.

the firing rate decreases at a constant value of injected current. In contrast, in the presence of CCh, the motion was variable; it could be essentially linear as shown in the somatic example in *Figure 1C2* or even counter-clockwise as in the dendritic example in *Figure 1G2*. The frequency/current (f/I) plots in *Figure 1—figure supplement 1* show the clockwise hysteresis in control and different cases of reversal or removal of adaptation in the presence of CCh for somatic recordings, with panels A and B showing acceleration, and panels C and D showing linearization. A linear motion indicates no hysteresis, although often the neurons fire at lower current injections on the down-ramp (defined as sustained linear in *Venugopal et al., 2015*), whereas counter-clockwise motion corresponds to an acceleration of the firing rate that would be observed during injection of a constant current. Both of these variants correspond to an effective removal of the spike rate adaptation by cholinergic activation. These results also demonstrate that cholinergic activity shifts the center of mass of firing to later times during a symmetric ramp. We speculate that the level of cholinergic modulation could therefore be a source of rapid modification of the timing of place cell firing relative to position in the place field. Our summary results in *Figure 1* show that carbachol similarly affects 2 s and 10 s ramps, and we present only the data for the 2 s ramps for subsequent figures.

Since the initial membrane potential and the amount of current injected could potentially affect the center of mass of firing, confounding the effects of cholinergic modulation, we plotted the adaptation index as a function of the initial membrane potential (*Figure 1—figure supplement 2A*) and of injected current amplitude (*Figure 1—figure supplement 2B*), in control conditions and in the presence of carbachol (2 µM) for each trial for all recorded neurons, both for somatic and dendritic recordings. Likewise, since the input resistance of the neuron could potentially skew the center of mass of firing, we plotted the average adaptation index as a function of the steady-state input resistance in control conditions and in the presence of carbachol for each somatic and dendritic recording (*Figure 1—figure supplement 2C*). In all cases, it is apparent that the main factor determining the adaptation index, and therefore whether the center of mass of firing occurs earlier or later in the ramp, is the presence or absence of carbachol rather than initial membrane potential, current injection amplitude, or input resistance.

The later onset of spiking in CCh in the experiments described above may be attributable to the smaller current ramps, but the persistence of spiking without adaptation is striking in comparison to control. The use of smaller ramps to achieve similar firing rates may be justified by analogy with the presynaptic action of ACh to reduce glutamate release (*Hasselmo, 2006*). Since the current injection protocols described above only account for the effects of cholinergic agonists on the postsynaptic neuron, we examined the effects of carbachol on synaptically-driven membrane potential ramps, where pre- and postsynaptic effects can be probed at the same time. To this end, we stimulated the Schaffer collateral fibers with bipolar electrodes and recorded the responses from the somata of CA1

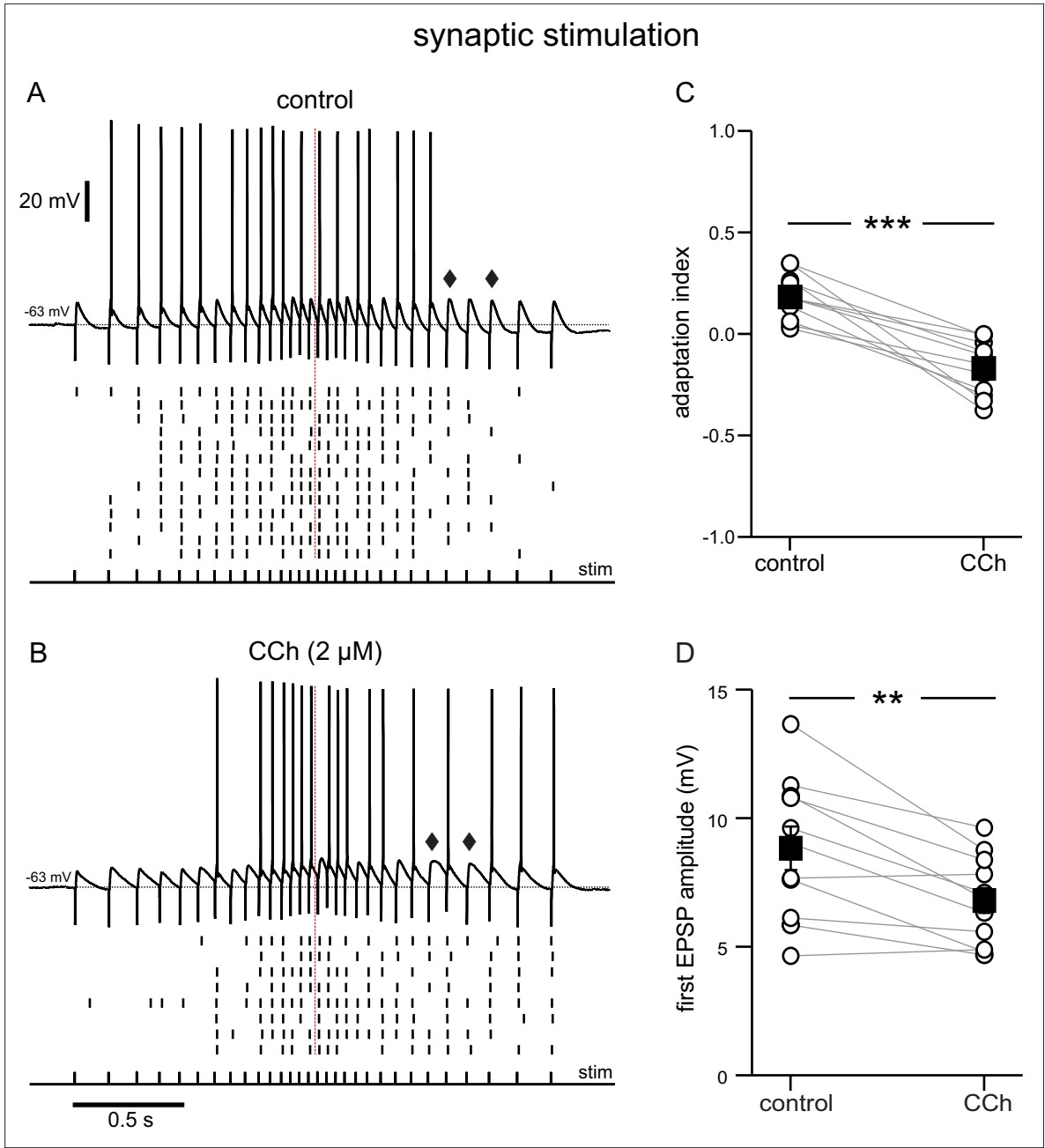

**Figure 2.** Cholinergic modulation of the response to electrical stimulation of Schaffer collaterals, recorded at the soma. (**A**) Top, synaptic response in control conditions to an input frequency that increases and decreases symmetrically, from 6.7 to 25 Hz, as denoted by the pulses, labeled stim. Bottom, Raster plot of multiple trials repeated at the same stimulation intensity. (**B**) Same as **A** during superfusion of 2 μM CCh. **A** and **B** are from the same neuron, and at the same stimulation intensity. Diamonds in **A** and **B** point to a longer EPSP decay time in the presence of CCh. (**C**) Summary data of the adaptation index for synaptic ramps, before and during CCh. (**D**) Summary data of the average EPSP amplitude measured during trials where the cells did not fire in response to the first stimulus pulse, before and during CCh. In both cases, open circles connected by gray lines represent indices for individual cells; black squares with error bars represent group averages ± SEM *** p<0.0005, ** p=0.002, paired t-test. Red dotted lines in the panels on the left mark the middle of the synaptic stimulation. Source data in "*Figure 2—source data 1*".

The online version of this article includes the following source data for figure 2:

**Source data 1.** Cholinergic modulation of the response to electrical stimulation of Schaffer collaterals, recorded at the soma.

pyramidal neurons (*Figure 2*). The electrical stimulation consisted of 30 stimuli for a total duration of ~2 s. The stimulation pattern was adjusted according to a linear, symmetric ramp as in *Hsu et al., 2018*, such that the frequency increased and decreased symmetrically, starting at 6.7 Hz and peaking in the middle of the ramp at 25 Hz, the target peak frequency in the current injection experiments. The intensity of stimulation was adjusted such that neurons would fire in response to 40–65% of the total inputs under control conditions, and was kept constant in control conditions and in the presence of carbachol. In both conditions, firing was more frequent in the middle of the ramp, because of the higher stimulation frequency. However, as in the current injection protocols in *Figure 1*, some variability was observed in the timing of the spikes across trials, as can be appreciated in the raster plots. In control conditions, the neurons tended to fire more in the first half of the ramp (*Figure 2A*), whereas in the presence of carbachol they tended to fire more on the second half of the ramp (*Figure 2B*). The adaptation index was calculated in a similar manner as in *Figure 1*, as the normalized difference between the number of spikes in the first and the second half of the ramp. On average, the adaptation index decreased from 0.18±0.03 in control to –0.17±0.04 in the presence of carbachol, t(10) = 6.873, p<0.0005, n=11 (*Figure 2C*). This shift was partly due to a decrease in the amplitude of the EPSPs (*Figure 2D*); the average amplitude of the first EPSP in control (8.83±0.82 mV) was significantly larger than the average first EPSP in CCh (6.81±0.52 mV; t(10) = 4.127; p=0.002; n=11). In addition, the decay of the EPSPs appeared to slow down later in the ramp in the presence of CCh (compare EPSPs indicated by diamonds in the two conditions), resulting in an increase in temporal summation and excitability. These results show that carbachol tends to shift the center of mass of firing to later in the ramp in the case of a synaptically driven depolarization as well as in response to a ramp of injected current.

Carbachol is often used to more easily study cholinergic signaling in biological tissues, because it is not susceptible to hydrolysis by acetylcholinesterase, which is preserved and active in hippocampal slices (*Avignone and Cherubini, 1999*). To verify that the effects of the endogenous neuromodulator were comparable to those of its synthetic analog, we repeated the experiments in *Figure 1* with different concentrations of acetylcholine. In this and all following sets of experiments, the amplitude of current injection was adjusted for each cell and condition to produce peak frequencies between 10 and 25 Hz as described above. Acetylcholine caused a depolarization of the $V_m$ in a dose-dependent manner. As before, we compensated for this depolarization by injecting hyperpolarizing current to restore the initial $V_m$ (*Figure 3*). Like carbachol, acetylcholine shifted the center of mass of firing to later in the ramp, albeit to a more moderate degree (*Figure 3A–D*). A one-way repeated measures ANOVA was carried out to determine if there was an effect on the adaptation index of three concentrations of ACh within the same neurons. The assumption of sphericity was violated, so a Greenhouse-Geisser correction was applied ($\varepsilon$=0.574). There was a significant effect of ACh on the adaptation index, plotted in *Figure 3E* (control 0.35±0.04; 2 µM ACh 0.14±0.04; 10 µM ACh 0.00±0.06; 15 µM ACh –0.02±0.06; F(1.722, 20.665)=44.360; p<0.0005, n=13). Bonferroni-adjusted post hoc tests demonstrated significant differences between the control condition and all concentrations of ACh (all comparisons p<0.0005), and between the lowest concentration and the subsequent higher concentrations (for 2 µM compared to 10 µM, p=0.001; for 2 µM compared to 15 µM, p=0.01). There was not a significant difference between 10 µM and 15 µM ACh. Due to likely degradation via acetylcholinesterase, we speculate that the actual concentration in the slices is lower than the applied one. CCh is not susceptible to degradation by acetylcholinesterase, and its concentration, which in our experiments is markedly lower than that used in many studies on cholinergic modulation of neuronal activity (*Bland et al., 1988*; *Fisahn et al., 1998*; *Knauer et al., 2013*), is therefore more consistent, which may explain the more pronounced shift in spiking in *Figure 1* as compared to *Figure 3*. Nonetheless, our results with ACh suggest that there are concentration-dependent effects of the endogenous neuromodulator on firing rate adaptation. Fluctuations in ACh levels could therefore have implications for the firing of a place cell within its place field.

## Cholinergic-associated shift in the timing of firing during the ramp is mediated by TRPM4 channels

An increase in firing on the down-ramp compared to the up-ramp induced by cholinergic activation could result from (1) a decrease in an outward current and/or (2) an increase in an inward current. The cascade of events that follow cholinergic activation in these neurons includes the suppression of a

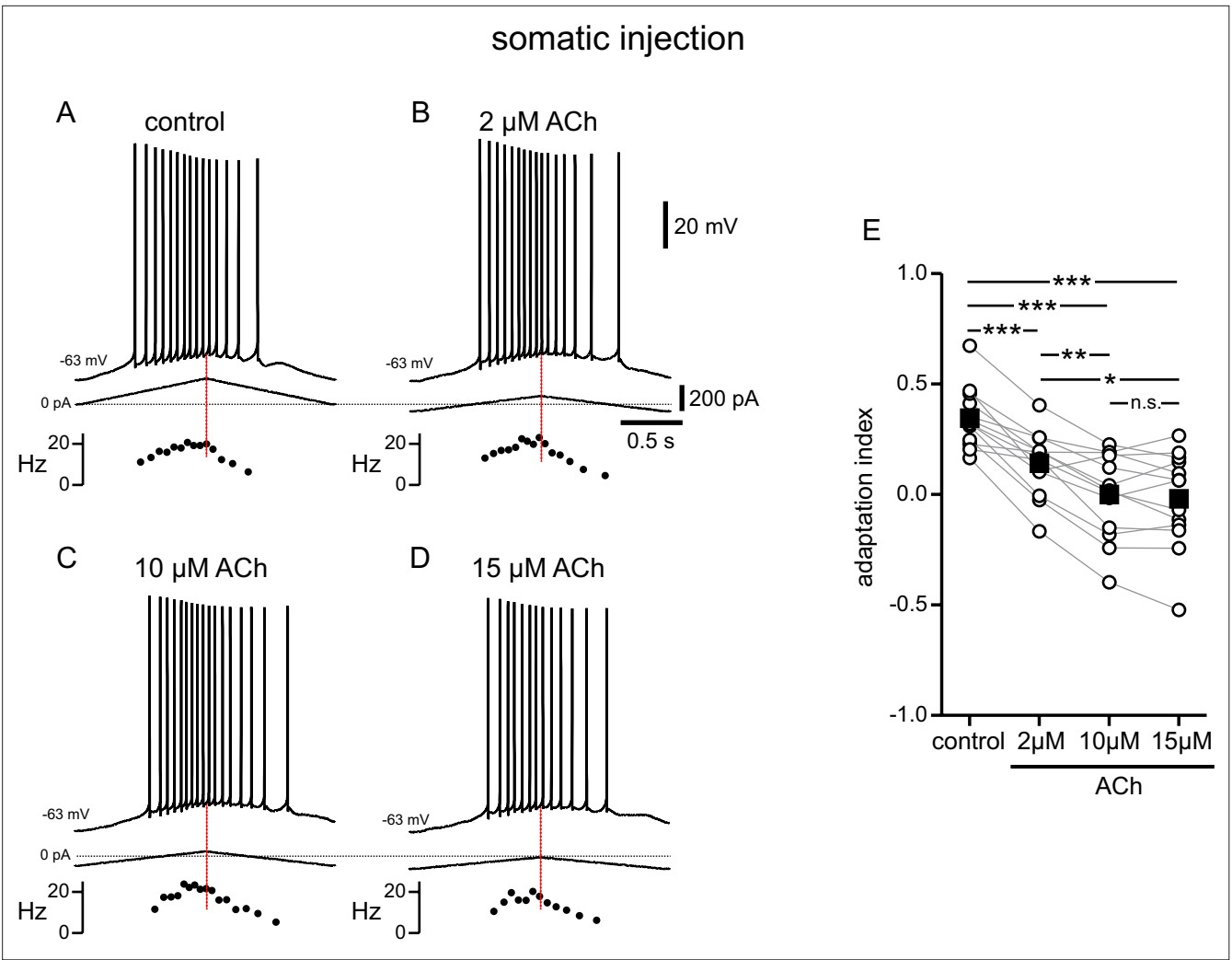

**Figure 3.** Modulation of responses to symmetric ramps by acetylcholine is concentration dependent. (**A**) Voltage trace recorded in the soma for a two second triangular current ramp injected in the soma in control. Current traces and instantaneous frequency plots are just below the voltage traces. (**B–D**) Same as A, during superfusion with 2 μM ACh (**B**), 10 μM ACh (**C**), and 15 μM ACh (**D**). A through D are from the same cell. (**E**) Summary data of the adaptation index for control and the three concentrations of ACh, applied in order of increasing concentration (n=13). Open circles connected by gray lines represent indices for individual cells over all four conditions. Black squares with error bars represent group averages ± SEM. *** p<0.0005, ** p=0.001, * p=0.01, one-way repeated measures ANOVA with a Greenhouse-Geisser correction and post hoc pairwise comparisons with a Bonferroni adjustment. Red dotted lines in the panels on the left mark the middle of the current injection ramp. Source data in "*Figure 3—source data 1*".

The online version of this article includes the following source data for figure 3:

**Source data 1.** Modulation of responses to symmetric ramps by acetylcholine is concentration dependent.

number of outward currents (*Halliwell and Adams, 1982*; *Hoffman and Johnston, 1998*; *Buchanan et al., 2010*; *Giessel and Sabatini, 2010*), some of which are also involved in firing rate adaptation (*Stocker et al., 1999*; *Gu et al., 2005*; *Pedarzani et al., 2005*; *Otto et al., 2006*; *Chen et al., 2014*). Although the SK channel clearly contributes to adaptation when the dendrites are stimulated at high frequency (>50 Hz; *Combe et al., 2018*), we have previously examined the contribution of small-conductance calcium-activated potassium (SK) current, M-type potassium current, and A-type potassium current with respect to the asymmetric firing in response to symmetric current ramps (*Upchurch et al., 2022*). We found that, in the relatively low range of firing frequencies evoked by this type of stimulation, the adaptation induced by these K⁺ currents (including SK) does not play a role in shifting action potentials to earlier times during the ramp. However, cholinergic activation has also been shown to activate a non-specific inward cationic current, $I_{CAN}$, that is thought to be carried by certain TRP channels (*Guinamard et al., 2010*; *Reboreda et al., 2011*). Flufenamic acid (FFA) has been

shown to block a variety of TRP channels in hippocampal neurons (*Partridge and Valenzuela, 2000*) and elsewhere (*Guinamard et al., 2013*). After observing the typical response to CCh, responses to symmetric ramps were recorded while FFA (100 µM) was applied along with CCh (*Figure 4A and B*). FFA shifted most action potentials back to the up-ramp as shown in the examples in *Figure 4A1–A3 and B1–B3*, effectively reversing the shift triggered by CCh. The differences in adaptation index across the three conditions were found to be significantly different by repeated measures ANOVA in both somatic recordings (control 0.35±0.05, CCh –0.36±0.09, CCh + FFA 0.25±0.06; $F_{(2,16)}$ = 65.835; p<0.0005; n=9; *Figure 4A4*) and dendritic recordings (control 0.52±0.02, CCh –0.54±0.11, CCh + FFA 0.20±0.08; $F_{(2,16)}$ = 60.680; p<0.0005; n=9; *Figure 4B4*). Bonferroni-adjusted pairwise comparisons indicated that for somatic recordings, the differences were significantly different between control and CCh (p<0.0005), and between CCh and CCh + FFA (p<0.0005), but not between control and CCh + FFA (p=0.43). For dendritic recordings, the differences were significantly different between control and CCh (p<0.0005), CCh and CCh + FFA (p<0.0005), and control and CCh + FFA (p=0.02). In most cases, the adaptation index reverted to a positive value with the application of FFA from a negative value in CCh alone, and in all cases was closer to the control value with FFA than with CCh alone. The failure of the adaptation index to return to control values in FFA may reflect an incomplete block of TRP channels or an additional effect of CCh. Overall, these data suggest that the shift in the center of mass of firing to later along the ramp that is associated with cholinergic activity is due, in large part, to activation of TRP channels.

The TRPC subfamily has been implicated as being responsible for $I_{CAN}$ in some neurons and is thought to be activated by elements downstream of muscarinic acetylcholine receptor activation (*Putney, 2005*), which makes it an attractive candidate, mechanistically, for this cholinergic-mediated shift in action potential firing along the depolarizing ramp. Cholinergic stimulation of $G_{q/11}$-coupled mAChRs results in phospholipase C hydrolysis of membrane constituent phosphatidylinositol 4,5-bisphosphate ($PIP_2$) into inositol 1,4,5-trisphosphate ($IP_3$) and diacylglycerol (DAG). $IP_3$ stimulates $IP_3$ receptor ($IP_3$R) associated $Ca^{2+}$ release from the endoplasmic reticulum, regulating TRPC channel activity via calmodulin (*Zhu, 2005*), while DAG activates TRPC channels directly (*Hofmann et al., 1999*; *Mederos Y Schnitzler et al., 2018*).

In a set of experiments similar to those above, the TRPC-specific antagonist SKF96365 (50 µM) was added after CCh was applied (*Figure 4C and D*). The differences in adaptation index across the three conditions were found to be significantly different by repeated measures ANOVA in somatic recordings (control 0.43±0.07, CCh –0.36±0.09, CCh + SKF –0.24±0.10; $F_{(2,14)}$ = 42.453; p<0.0005; n=8; *Figure 4C4*) and dendritic recordings (control 0.47±0.11, CCh –0.24±0.07, CCh + SKF –0.23±0.06; $F_{(2,14)}$ = 118.541; p<0.0005; n=8; *Figure 4D4*). However, Bonferroni-adjusted pairwise comparisons demonstrated that the significant differences lie both between control and CCh (soma p<0.0005; dendrite p<0.0005) and control and CCh + SKF (soma p=0.001; dendrite p<0.0005), not between CCh and CCh + SKF (soma p=0.68; dendrite p=1.00). Thus, SKF96365 did not significantly shift the adaptation index in the same manner as did FFA, suggesting that the TRP channels involved in the CCh-mediated forward shift are not of the TRPC subtype.

TRPM4 channels have likewise been implicated in downstream effects associated with $G_q$-coupled signaling (*Launay et al., 2002*; *Dutta Banik et al., 2018*). TRPM4 channels are activated by $Ca^{2+}$ binding to the cytoplasmic side combined with depolarization (*Nilius et al., 2003*; *Autzen et al., 2018*), but these channels are impermeable to $Ca^{2+}$. Instead, TRPM4 channels facilitate a rise in $[Ca^{2+}]_i$ indirectly, by depolarizing the membrane potential (*Launay et al., 2002*) and therefore activating voltage-gated $Ca^{2+}$ channels (*Li et al., 2021*). Cationic current through these channels and the resulting $[Ca^{2+}]_i$ elevation has been implicated in bursting, persistent firing, and oscillatory activity in some neurons (*Lin et al., 2017*; *O'Malley et al., 2020*; *Li et al., 2021*). In order to investigate the involvement of TRPM4 channels in the cholinergic-mediated shift in adaptation index, the TRPM4 channel antagonist CBA (50 µM) was added after recording the response to CCh (*Figure 5A and B*). Repeated measures ANOVA showed significant differences in the adaptation indices in somatic recordings (control 0.29±0.05, CCh –0.37±0.10, CCh + CBA 0.17±0.07; $F_{(2,16)}$ = 66.257; p<0.0005; n=9; *Figure 5A4*) and dendritic recordings (control 0.39±0.07, CCh –0.41±0.10, CCh + CBA 0.08±0.05; $F_{(2,16)}$ = 34.848; p<0.0005; n=9; *Figure 5B4*). Bonferroni-corrected pairwise comparisons indicated that for somatic recordings, the differences were significant between control and CCh (p<0.0005), and between CCh and CCh + CBA (p<0.0005), but not between control and CCh + CBA (p=0.16).

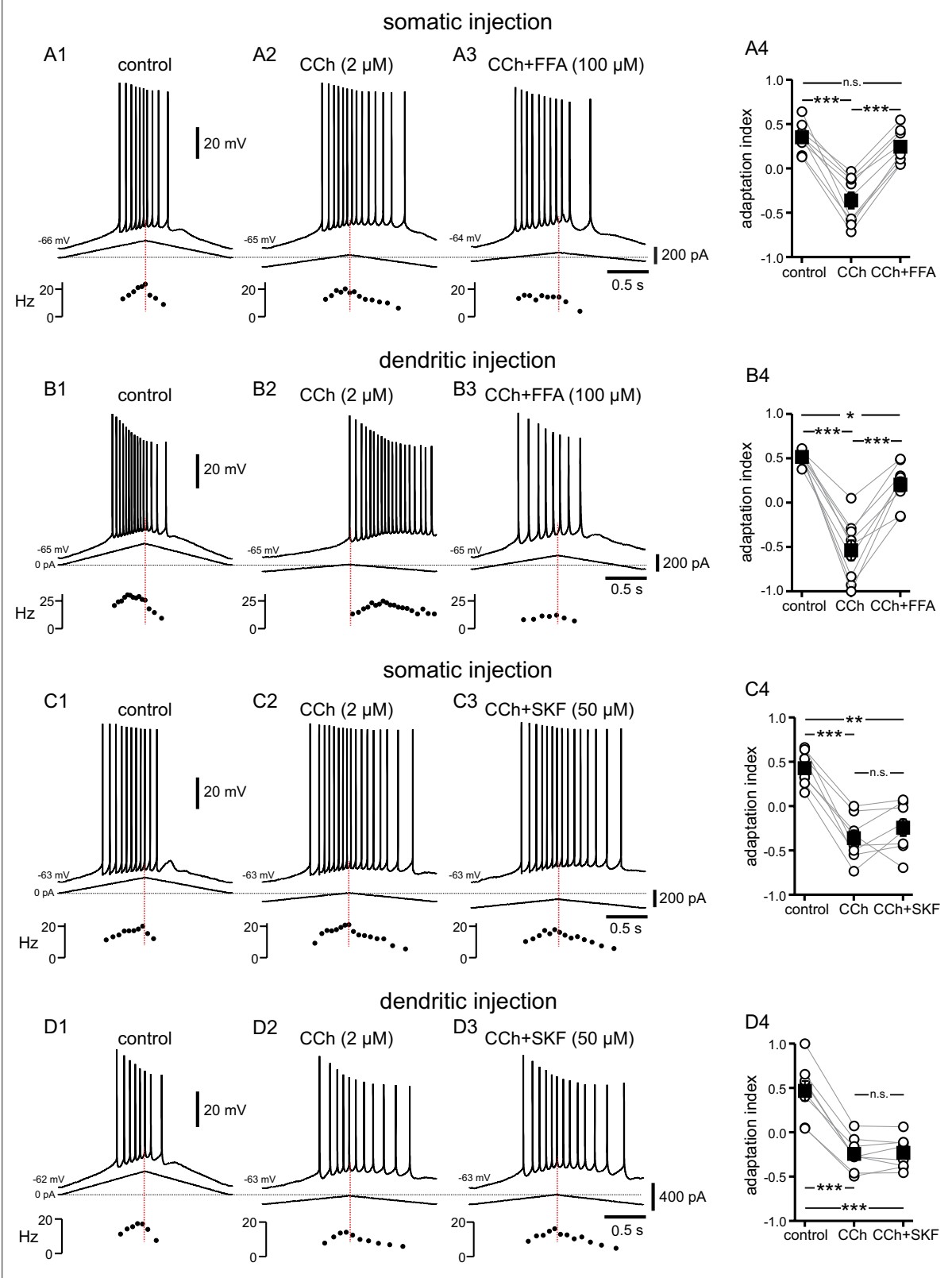

**Figure 4.** Cholinergic-associated shift in the timing of firing on ramp is mediated by TRP channels, but notably not TRPC channels. (**A**) Somatic recordings for a two second triangular current ramp injected in the soma in control (**A1**), CCh (2 μM) (**A2**), and CCh + FFA (100 μM) (**A3**) applied subsequently to the same cell. Current traces and instantaneous frequency plots are just below the voltage traces. Summary data (**A4**) of the adaptation index for control, CCh, and CCh + FFA (n=9). (**B**) Same as in A, but for dendritic injection and recordings. Summary data (**B4**) of the adaptation index

*Figure 4 continued on next page*

*Figure 4 continued*

for control, CCh, and CCh + FFA (n=9). Open circles connected by gray lines represent indices for individual cells, over all three treatments. Black squares with error bars represent group averages ± SEM. (**C**) Somatic recordings for a two second triangular current ramp injected in the soma in control (**C1**), CCh (2 µM) (**C2**), and CCh + SKF96365 (50 µM) (**C3**) applied subsequently to the same cell. Current traces and instantaneous frequency plots are just below voltage traces. Red dotted line marks the middle of the ramp. Summary data (**C4**) of the adaptation index for control, CCh, and CCh + SKF96365 (n=8). (**D**) Same as in **C**, but for dendritic injection and recordings. Summary data (**D4**) of the adaptation index for control, CCh, and CCh + SKF96365 (n=8). *** p<0.0005, ** p=0.001, * p=0.02, n.s. not significant, one-way repeated measures ANOVA and post hoc pairwise comparisons with a Bonferroni adjustment. Source data in "*Figure 4—source data 1*".

The online version of this article includes the following source data for figure 4:

**Source data 1.** Cholinergic-associated shift in the timing of firing on ramp is mediated by TRP channels, but notably not TRPC channels.

For dendritic recordings, Bonferroni-corrected pairwise comparisons showed that the differences were significant between control and CCh (p<0.0005), between CCh and CCh + CBA (p=0.002), and between control and CCh + CBA (p=0.02). In most cells, the adaptation index became negative in CCh and reverted to a positive value with the addition of CBA, and in all cases the adaptation index was higher in CCh + CBA than in CCh alone.

In order to corroborate that the effect of CBA is due to block of TRPM4 channels, we investigated the effect of a second TRPM4 channel antagonist, 9-phenanthrol (9Ph), added to CCh in somatic recordings. *Figure 5C* illustrates the shift of most of the spiking activity towards the up-ramp, moving the adaptation index in the positive direction, when 9-phenanthrol (100 µM) was applied on top of CCh. The differences in adaptation index between groups are significant, as demonstrated by repeated measures ANOVA (control 0.40±0.03, CCh –0.14±0.06, CCh + 9Ph 0.19±0.06; $F_{(2,14)}$ = 57.640; p<0.0005; n=8; *Figure 5C4*). Bonferroni-corrected pairwise comparisons indicated that the differences were significant between control and CCh (p<0.0005), between CCh and CCh + 9Ph (p=0.001), and between control and CCh + 9Ph (p=0.03).

Taken together, these data with the non-selective TRP channel blocker FFA, the TRPC channel blocker SKF96365, and two different TRPM4 channel antagonists suggest that the shift in adaptation index that we see with CCh is largely mediated through activation of TRPM4, rather than TRPC, channels.

## Does $Ca^{2+}$ release from internal sources activate TRPM4 channels?

We next tried to determine the source of $Ca^{2+}$ required to activate TRPM4 channels. We first focused on $IP_3$ receptors ($IP_3Rs$), which are activated through mAChRs and $G_q$ as described above, using the antagonist Xestospongin C (*Gafni et al., 1997*) in the intracellular solution (*Figure 6A*). Since the antagonists rapidly dialyzed into the cytosol, in this figure control data reflect the effect of the antagonists before the addition of CCh. We initially used Xestospongin C at 1 µM, which was previously found to be effective in inhibiting TRPM4 channel activation via mGluRs in neurons of the preBötzinger nucleus (*Pace et al., 2007*). Since this concentration was not effective in preventing the cholinergic shift in the center of mass of firing in our experiments, in six out of nine experiments we increased the concentration to 2 µM. Neither concentration of intracellular Xestospongin C appeared to change the firing features in control conditions. The shift induced by carbachol did not appear to depend on the concentration of Xestospongin C (*Figure 6A3*), therefore we averaged the data in these two conditions. The adaptation index still decreased significantly ($t(8)$ = 10.574, p<0.0005; n=9, as per a paired t-test) from 0.31±0.02 in control conditions to –0.38±0.07 in the presence of CCh (2 µM). This result argues strongly against the hypothesis that the effects of the activation of muscarinic receptors on the shift of the center of mass of firing are mediated by activation of $IP_3Rs$. Ryanodine receptors (RyRs) are also expressed on the endoplasmic reticulum in CA1 pyramidal neurons, and appear to contribute selectively to spine $Ca^{2+}$ elevations during LTP at the CA3-CA1 synapses (*Raymond and Redman, 2006*). We therefore tested the hypothesis that the increase of $[Ca^{2+}]_i$ in a nanodomain leading to TRPM4 channel activation could be due to $Ca^{2+}$-induced $Ca^{2+}$ release (CICR) through RyRs following $Ca^{2+}$ influx through voltage-dependent $Ca^{2+}$ channels. In these experiments, we included ryanodine at 40 µM, a concentration known to inhibit RyRs (*Ehrlich et al., 1994*; *Isokawa and Alger, 2006*), in the intracellular solution. Notably, one out of nine of the neurons recorded from in these conditions had a slightly negative adaptation index already under control conditions (*Figure 6B3*) when RyR were inhibited. Moreover, three more neurons appeared to fire irregularly, with some doublets, under

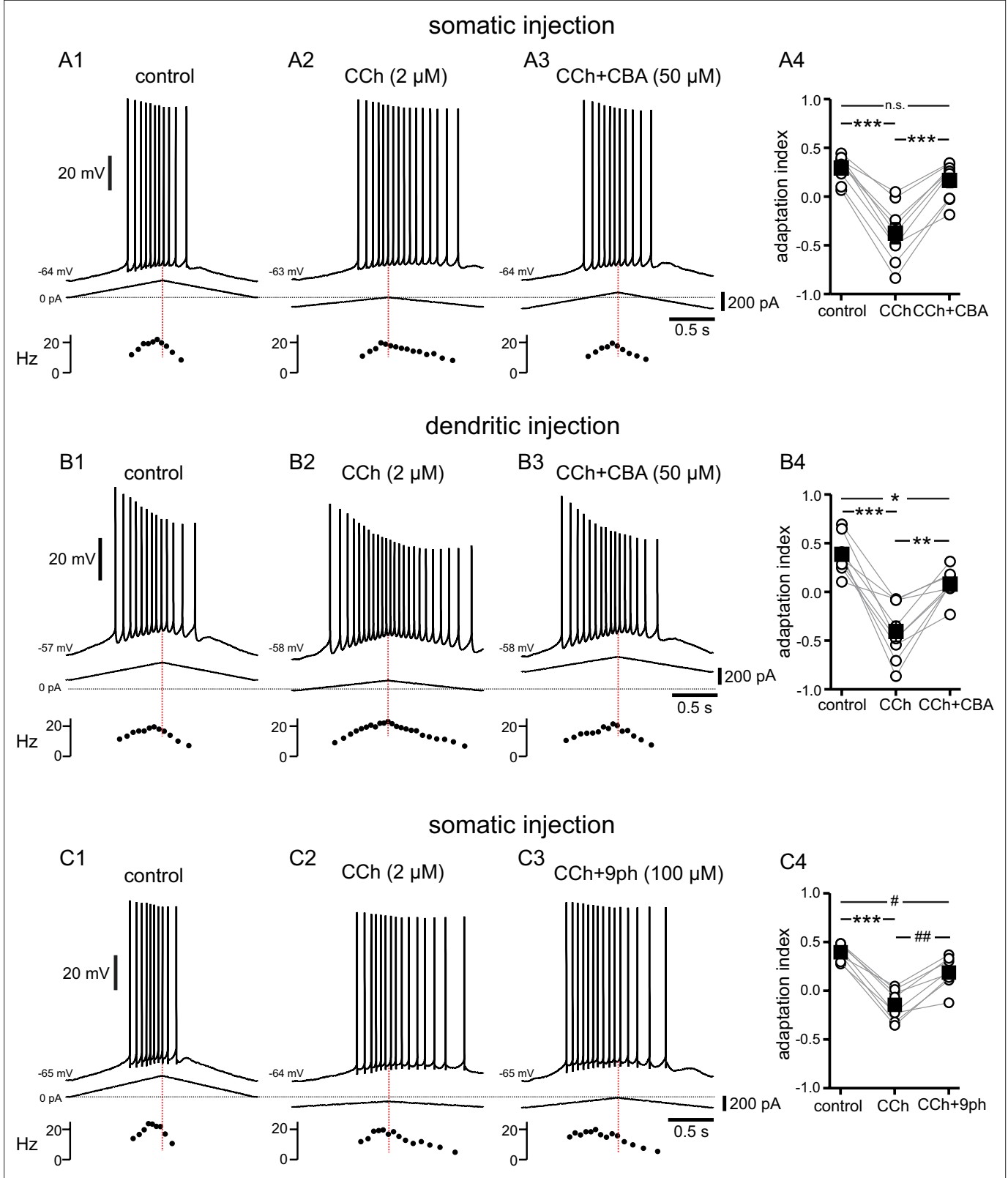

**Figure 5.** Cholinergic-mediated shift in the center of mass of firing is mediated by TRPM4 channels. (**A**) Somatic recordings for a two second triangular current ramp injected in the soma in control (**A1**), CCh (2 μM) (**A2**), and CCh + CBA (50 μM) (**A3**) applied subsequently to the same cell. Current traces and instantaneous frequency plots are just below voltage traces. Red dotted line marks the middle of the ramp. Summary data (**A4**) of the adaptation index for control, CCh, and CCh + CBA (n=9). Open circles connected by gray lines represent indices for individual cells, over all three treatments.

*Figure 5 continued on next page*

Figure 5 continued

Black squares with error bars represent group averages ± SEM. (**B**) Same as in A, but for dendritic injection and recordings. Summary data (**B4**) of the adaptation index for control, CCh, and CCh + CBA (n=9). (**C**) Same as A, but for control (**C1**), CCh (2 µM) (**C2**), and CCh + 9-phenanthrol (100 µM) (**C3**) applied subsequently to the same cell. Summary data (**C4**) of the adaptation index for control, CCh, and CCh + 9-phenanthrol (n=8). *** p<0.0005, **p=0.002, * p=0.02, ## p=0.001, # p=0.03, n.s. not significant, one-way repeated measures ANOVA and post hoc pairwise comparisons with a Bonferroni adjustment. Source data in "*Figure 5—source data 1*".

The online version of this article includes the following source data for figure 5:

**Source data 1.** Cholinergic-associated shift in the timing of firing on ramp is mediated by TRP channels, but notably not TRPC channels.

control conditions (an example is shown in *Figure 6C*), a feature that was not observed in other experimental conditions. This change could be due to the close juxtaposition of RyRs in the endoplasmic reticulum and various K+ channels on the membrane of CA1 neurons (*Vierra et al., 2019*; *Tedoldi et al., 2020*; *Sahu and Turner, 2021*). Upon addition of CCh in the bath, the adaptation index became significantly more negative in all nine neurons recorded with intracellular ryanodine, as per a paired t-test (control 0.28±0.05, CCh –0.46±0.07; t(8) = 11.99; p<0.0005; n=9), suggesting that RyRs also do not play a role in the cholinergic shift of the center of mass of firing.

## Multicompartmental modeling of cholinergic modulation of asymmetric firing responses

We next turned to our detailed multicompartmental model, in order to help identify the underlying mechanisms that contribute to the cholinergic modulation that we observe experimentally in vitro. The model was implemented in a reconstructed morphology (*Megías et al., 2001*) with 144 compartments, each of which can be represented with an equivalent circuit with heterogeneous channel conductance densities and kinetics as described in the Methods. Previously (*Upchurch et al., 2022*), this model was used to capture the firing rate adaptation observed under control conditions as shown in *Figure 7A1* for short triangular ramps of injected current. The model performs a material balance on $Ca^{2+}$ ions to provide estimated free $Ca^{2+}$ concentrations in concentric shells beneath the membrane, as shown in the top panels of *Figure 7*. The control simulation for a 2 s somatic ramp in *Figure 7A1* captures the experimentally observed decrease in spike number and frequency on the down-ramp as compared to the up-ramp (adaptation index = 0.28), as well as a slight decrease in spike amplitude, previously shown to be due to a decrease in available $Na_V$ channels (*Upchurch et al., 2022*). The plot of instantaneous frequency versus level of injected current at the right captures the clockwise pattern observed in *Figure 1C1 and G1* corresponding to firing rate adaptation. *Figure 7A2* shows the $[Ca^{2+}]$ in the outer shell of the soma in nM, *Figure 7A3* replots this same concentration in µM (the nanodomain in this case is contiguous with the outer shell so it is not actually a nanodomain), and *Figure 7A4* shows that this concentration is insufficient to activate TRPM4 channels. In order to model the effect of CCh, we added a nanodomain for the spatially restricted $Ca^{2+}$ concentration which we hypothesize is sensed by the $Ca^{2+}$ binding site on TRPM4 channels; also note that, as in the experiments, the amount of injected current is lower for the simulated CCh. We showed in *Figure 6* that TRPM4 channel activation is not affected by antagonists of $IP_3Rs$ and RyRs, excluding a contribution of $Ca^{2+}$ release from the ER. Although the source of the $Ca^{2+}$ increase that activates TRPM4 channels is currently unknown, it likely depends in some way upon the activation of voltage-gated $Ca^{2+}$ channels during the ramp. We therefore made the $Ca^{2+}$ influx in the nanodomain proportional to the total instantaneous $Ca^{2+}$ current across the membrane in the model, but only during simulated activation of the muscarinic receptor. Adding the $Ca^{2+}$ influx from an unknown source to a nanodomain in order to simulate bath application of CCh (*Figure 7B1*) replicates the experimentally observed shift in adaptation index, with more spikes occurring on the down-ramp as compared to the up-ramp (adaptation index = –0.29). The plot of instantaneous frequency versus injected current captures the more extreme observed effect of CCh as illustrated in *Figure 1G2*; the motion is counter-clockwise corresponding to the acceleration of the firing rate. *Figure 7B2* shows the $Ca^{2+}$ concentration in the outer shell of the soma in nM, *Figure 7B3* shows the $Ca^{2+}$ concentration in µM in the hypothesized nanodomain, and *Figure 7B4* shows that this concentration is sufficient to activate TRPM4 channels.

*Figure 8* shows why spike rate adaptation is not observed in the simulated presence of CCh. *Figure 8A* repeats the voltage and current traces from the previous figure, except that the simulated CCh traces (magenta) are superimposed over the control traces (black). Our model includes a four

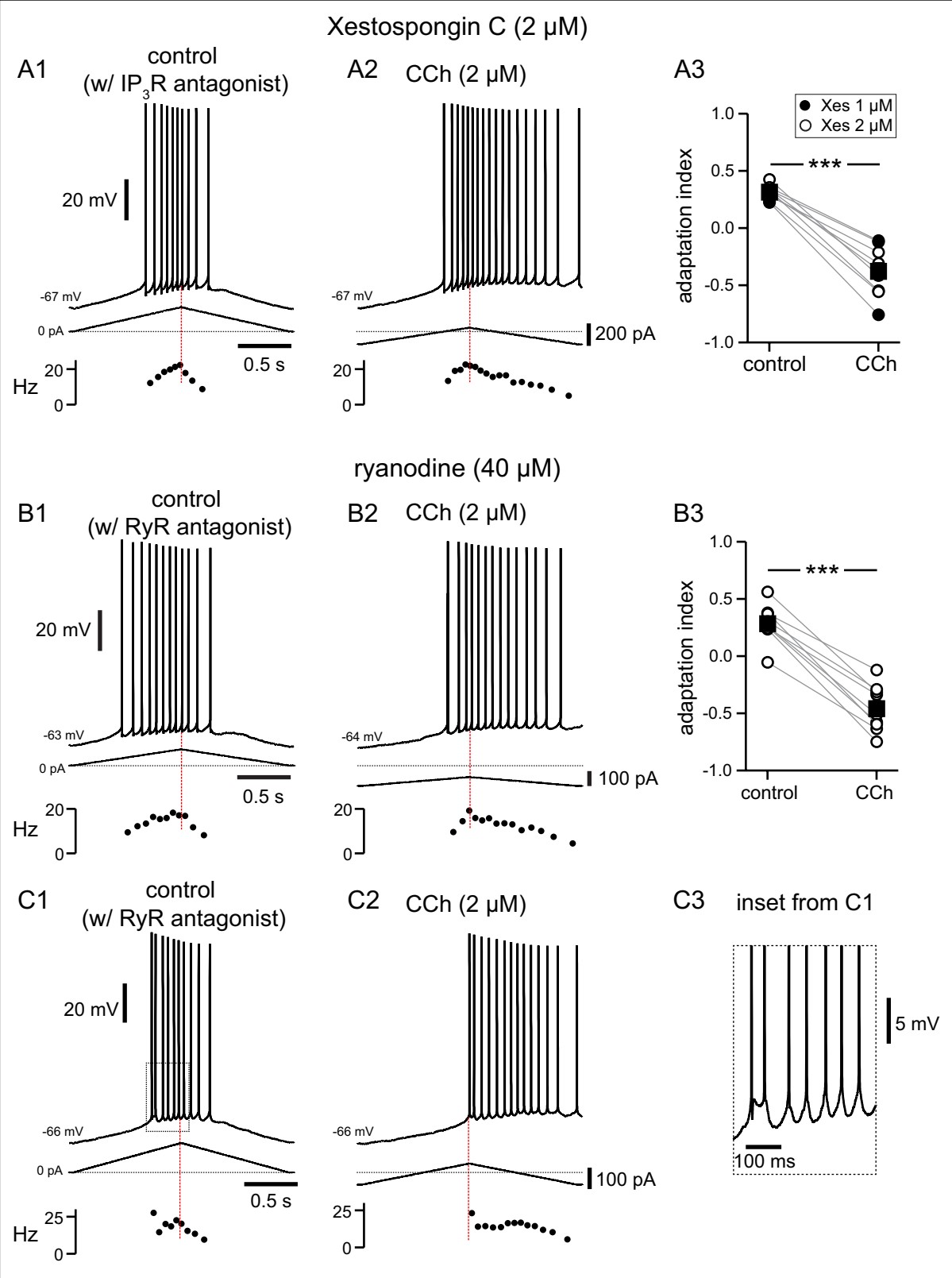

**Figure 6.** Pharmacological block of IP₃Rs or RyRs has no effect on CCh-mediated shift in firing along ramp. (**A**) Somatic recordings during intracellular dialysis with Xestospongin C (2 μM), for a two second triangular current ramp injected in the soma before (**A1**), and during superfusion with CCh (2 μM) (**A2**). Current traces and instantaneous frequency plots are just below voltage traces. Summary data (**A3**) of the adaptation index for Xestospongin C (1–2 μM) and CCh (n=9). Circles connected by gray lines represent ratios for individual cells, before and during CCh. Filled circles represent cells

*Figure 6 continued on next page*

*Figure 6 continued*

dialyzed with 1 μM Xestospongin C and open circles represent cells dialyzed with 2 μM Xestospongin C. Black squares with error bars represent group averages ± SEM. (**B**) Same as A, but intracellular dialysis is with ryanodine (40 μM). Summary data (**B3**) of the adaptation index for ryanodine and CCh (n=9). Individual cells represented by open circles. *** p<0.0005, paired t-test. In both sets of experiments, control data reflect the effect of either IP₃R or RyR antagonist before the addition of CCh, because the dialysis started to be apparent quickly after break-in. Red dotted lines in the left panels mark the middle of the ramp. C1 and C2. Same as B1 and B2; C3 is an expanded version of the inset in C1. Source data in "*Figure 6—source data 1*".

The online version of this article includes the following source data for figure 6:

**Source data 1.** Pharmacological block of IP3Rs or RyRs has no effect on CCh-mediated shift in firing along ramp.

state Markov model of the Na$_V$ channel (see Materials and methods). Occupancy in the open state between action potentials is the substrate for the persistent Na$_V$ current during the interspike interval. *Figure 8B* shows that the occupancy in the open state of the Markov model of the Na$_V$ channel in the soma is not greatly different between the two conditions, except for excursions produced by spiking, and *Figure 8D* shows the same result for the somatic Na$_V$ current. Our previous work (*Upchurch et al., 2022*) attributed spike rate adaptation to reduced persistent Na$_V$ current due to increased long-term inactivation. *Figure 8C* shows that although long-term inactivation of the Na$_V$ channel (occupancy in the I2 state) starts later in simulated CCh, it eventually progresses to higher levels than in control because of the increased number of spikes. Nonetheless, no spike rate adaption was observed in simulated CCh in *Figure 7B1* because the TRPM4 current is about 4-fold larger than the persistent Na$_V$ current (compare *Figure 8D and E*). *Figure 8F* shows that the additional TRPM4 current more than compensates for the smaller level of injected ramp current; as a consequence, the net inward current is comparable on the up-ramp and larger in magnitude during the down-ramp for the simulated CCh case than control. This causes spiking to persist longer despite higher levels of Na$_V$ channel inactivation.

When activated by Ca$^{2+}$, TRPM4 channels cause a depolarization and additional spiking that further increases Ca$^{2+}$ influx resulting in a positive feedback loop, with the regenerative inward TRPM4 current contributing to action potential initiation. *Figure 9* elaborates on the mechanisms by which Ca$^{2+}$ and voltage differentially contribute to TRPM4 channel activation in removing or reversing spike frequency adaptation. It repeats the traces in control (black) and simulated CCh (magenta) from *Figure 7* on an expanded scale and adds two conditions as described below. The insets to the right in *Figure 9A* show f/I curves that capture the qualitative features of the f/I curves shown in *Figure 1C1, C2, and G1 and G2*. Qualitatively, in control (black traces), there are more spikes on the up-ramp than on the down-ramp, and the instantaneous frequency is lower on the down-ramp for similar values of injected current during the up-ramp. The contribution of the increase in Ca$^{2+}$ concentration in the nanodomain was examined by turning off the voltage-dependence of the TRPM4 channel (light blue traces). In this case, the voltage component of TRPM4 channel activation was set to its steady state value at the resting V$_m$ (see Materials and methods).

Simply adding the nanodomain accomplishes the shift in center of mass (compare blue trace in *Figure 9A* to control trace in black) and removes the adaptation, with a very slight tendency to accelerate the firing. However, without the contribution of the voltage-dependence of TRPM4 channels, the resulting Ca$^{2+}$ influx is lower than in the original simulation of the effects of CCh (compare the blue traces to the magenta traces in *Figure 9B and C*), resulting in less TRPM4 channel activation (compare the blue trace to the magenta trace in *Figure 9D*) and therefore fewer total spikes. To determine the contribution of the positive feedback loop by which Ca$^{2+}$ entry recruits more Ca$^{2+}$ entry via additional spiking and depolarization, we recorded the Ca$^{2+}$ traces in each nanodomain (one for each segment of every section in NEURON, see Methods) of the original CCh simulation (magenta trace in *Figure 9C*). We then used this Ca$^{2+}$ trace as the [Ca$^{2+}$] in the nanodomain during simulations with the voltage dependence removed as described above. The purple trace in *Figure 9A* shows that the positive feedback loop added two spikes (16 in the purple trace versus 14 in the blue trace) by slightly increasing the frequency during the spike train and allowing spiking to continue longer. The f/I plot shows a little more acceleration of the firing rate. The purple trace in *Figure 9B* shows that [Ca$^{2+}$] in the outermost shell is slightly different than both the blue and magenta traces, reflecting the different numbers of spikes. Note that the purple and magenta traces in *Figure 9C* are identical because the magenta trace for [Ca$^{2+}$] in the nanodomain (original CCh simulation) was input to the nanodomain in the simulations shown

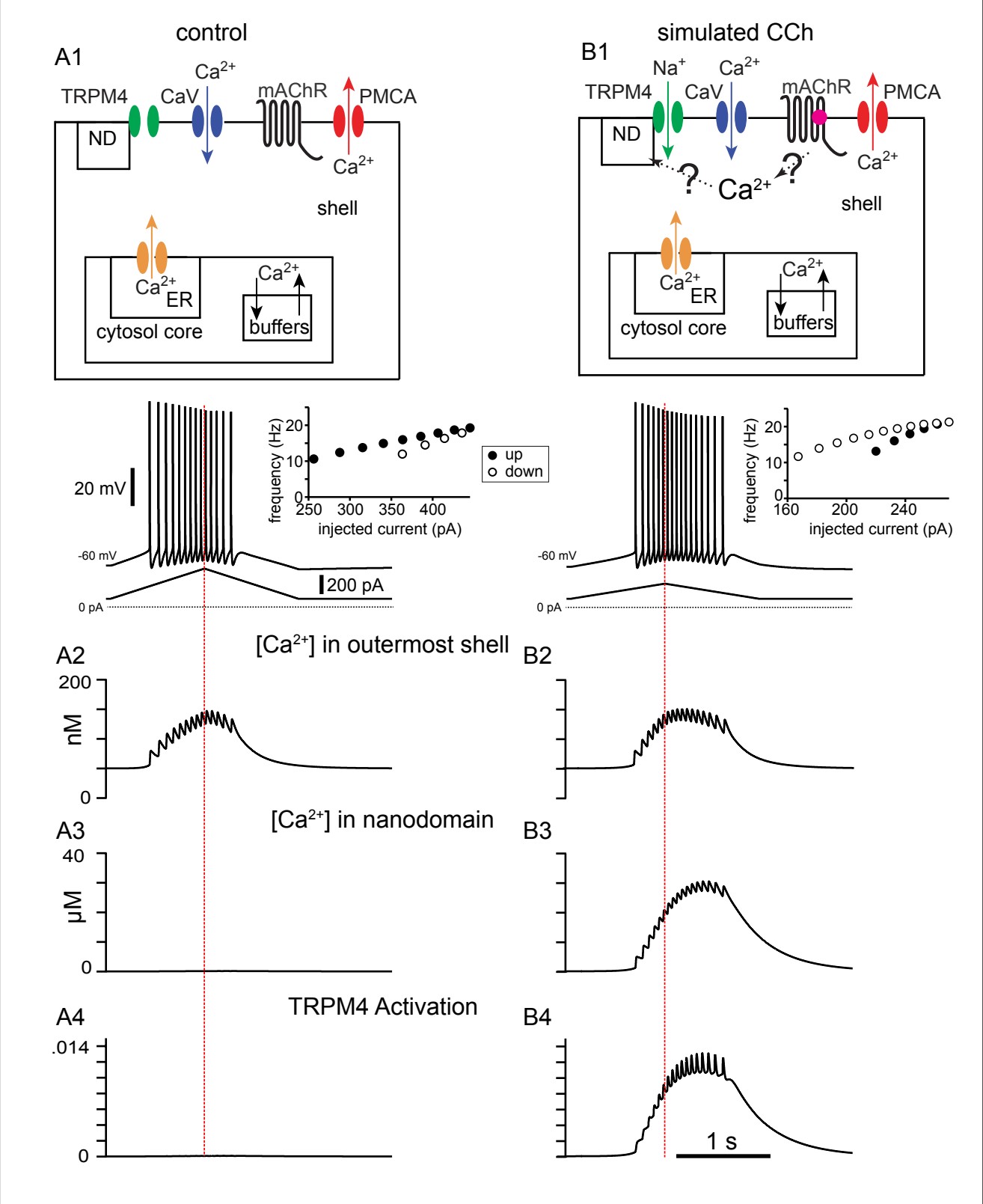

**Figure 7.** Simulated voltage responses to triangular current ramp injections, and underlying mechanisms. (**A**) Simulated control response to triangular current ramp. (**A1**) Top. Simplified model calcium handling schematic. For simplicity, only one annulus (shell) is shown, and the rest are represented as the core. Ca²⁺ enters via voltage-gated Ca²⁺ channels (CaV) and is removed by the plasma membrane Ca²⁺ ATPase (PMCA) pump. Inside the cell, buffering and the endoplasmic reticulum (ER) are also modeled. The TRPM4 channel is not permeable to Ca²⁺ and senses [Ca²⁺] in a nanodomain

*Figure 7 continued on next page*

*Figure 7 continued*

(ND). In the control simulation, the nanodomain is contiguous with the cytosol. Bottom trace shows the voltage responses, plotted in the somatic compartment, to a triangular current ramp applied in the soma (shown below). (**A2**) $Ca^{2+}$ concentration (in nM) in the outermost shell of the somatic compartment. (**A3**) The nanodomain $Ca^{2+}$ concentration (same as the that of the outer shell in A2 in this case) is negligible when plotted in µM scale. (**A4**) TRPM4 activation is negligible in control. B. Simulated response to triangular current ramp during simulated application of CCh. (**B1**) As in A1 except for ligand (magenta dot) binding to the metabotropic acetylcholine receptor (mAChR), triggering $Ca^{2+}$ influx into a nanodomain linked to the TRPM4 channel. The voltage trace shows that the center of mass of firing is shifted to later times. (**B2**) $Ca^{2+}$ concentration (in nM) in the outermost shell of the somatic compartment. (**B3**) In contrast to control, the addition of a nanodomain with privileged $Ca^{2+}$ access allows micromolar concentrations to be reached and sensed by TRPM4 channels. (**B4**) In contrast to control, TRPM4 is now activated during the ramp. Vertical dashed red line shows the peak of the current ramp.

in purple in the other panels. The additional TRPM4 channel activation (compare purple to blue trace in *Figure 9D*) results from the increased $[Ca^{2+}]$ in the nanodomain. Adding the full voltage-dependence to the TRPM4 channel slightly raises the overall level of TRPM4 channel activation. The voltage-dependence also allows for fast excursions in channel activation that could contribute to action potential initiation (compare magenta trace in *Figure 9D* to the blue and purple ones). The voltage-dependence contributes one additional spike (17 in the magenta trace versus 16 in the purple trace in *Figure 9A*) and accentuates spike rate acceleration due to the positive feedback loop, as shown in the f/I curves.

## Evidence for tight coupling between TRPM4 channels and the calcium source that activates them

In order to test experimentally the model prediction that the highly localized $Ca^{2+}$ elevation in a nanodomain next to the TRPM4 channels gives rise to the CCh-mediated shift in the center of mass of firing, the calcium chelator BAPTA was included in the recording pipette, and somatic recordings obtained before and during addition of CCh. In *Figure 10*, control data reflect the effect of BAPTA before the addition of CCh, rather than control data such as in previous figures, because the dialysis occurred quickly. As in previous sets of experiments, the amount of current injected was chosen in order to achieve a target firing frequency of 10–25 Hz. With intracellular dialysis of BAPTA, the current needed to reach these frequencies was consistently higher in these experiments (compare scale bars of injected current to those in previous figures). In addition, intracellular dialysis with BAPTA tended to shift the firing earlier in the ramp, with fewer spikes on the down-ramp, therefore making the adaptation index closer to 1. In *Figure 10A*, 10 mM BAPTA was included in the internal solution. As previously noted, the addition of CCh in the bath caused a depolarization that was compensated for by the injecting hyperpolarizing current. Moreover, despite the presence of BAPTA, the adaptation index became significantly smaller in the presence of CCh, as per a paired t-test (control 0.82±0.05, CCh –0.12±0.11; t(9) = 8.594; p<0.0005; n=10; *Figure 10A3*); therefore, 10 mM BAPTA did not prevent the cholinergic-mediated shift. It has been shown that a higher concentration of BAPTA may be required to prevent TRPM4 channel activation in neurons of the preBötzinger nucleus (*Pace et al., 2007*; *Picardo et al., 2019*), therefore, in *Figure 10B*, BAPTA in the internal solution was increased to 30 mM. With a free $[Ca^{2+}]_i$ of 100 nM (see Materials and methods), CCh never produced negative adaptation indices that were seen in previous figures, although the $V_m$ did depolarize and was restored by the injection of tonic hyperpolarizing current. CCh still had a significant effect on the adaptation index, as per a paired t-test (control 0.62±0.05, CCh 0.37±0.06; t(7) = 5.069; p=0.001; n=8; *Figure 10B3*). However, when the free $[Ca^{2+}]_i$ was adjusted to 10 nM in the presence of intracellular 30 mM BAPTA (*Figure 10C*), the adaptation index did not change significantly when CCh was applied in the bath (control 0.73±0.03, CCh 0.76±0.03; t(11) = 0.776; p=0.45; n=12; *Figure 10C3*). In addition, the depolarization normally observed in the presence of CCh did not occur in these conditions, and the amount of injected current was comparable to that injected in control. These data demonstrate that a concentration of BAPTA higher than the typical value (10 mM) used to disrupt $Ca^{2+}$-dependent mechanisms (*Harney et al., 2006*) is required to prevent TRPM4 channel activation, and that free $[Ca^{2+}]_i$ of <100 nM is necessary to disrupt TRPM4 channel activation. Together, these data suggest a close physical relationship between the source of $Ca^{2+}$ and TRPM4 channels.

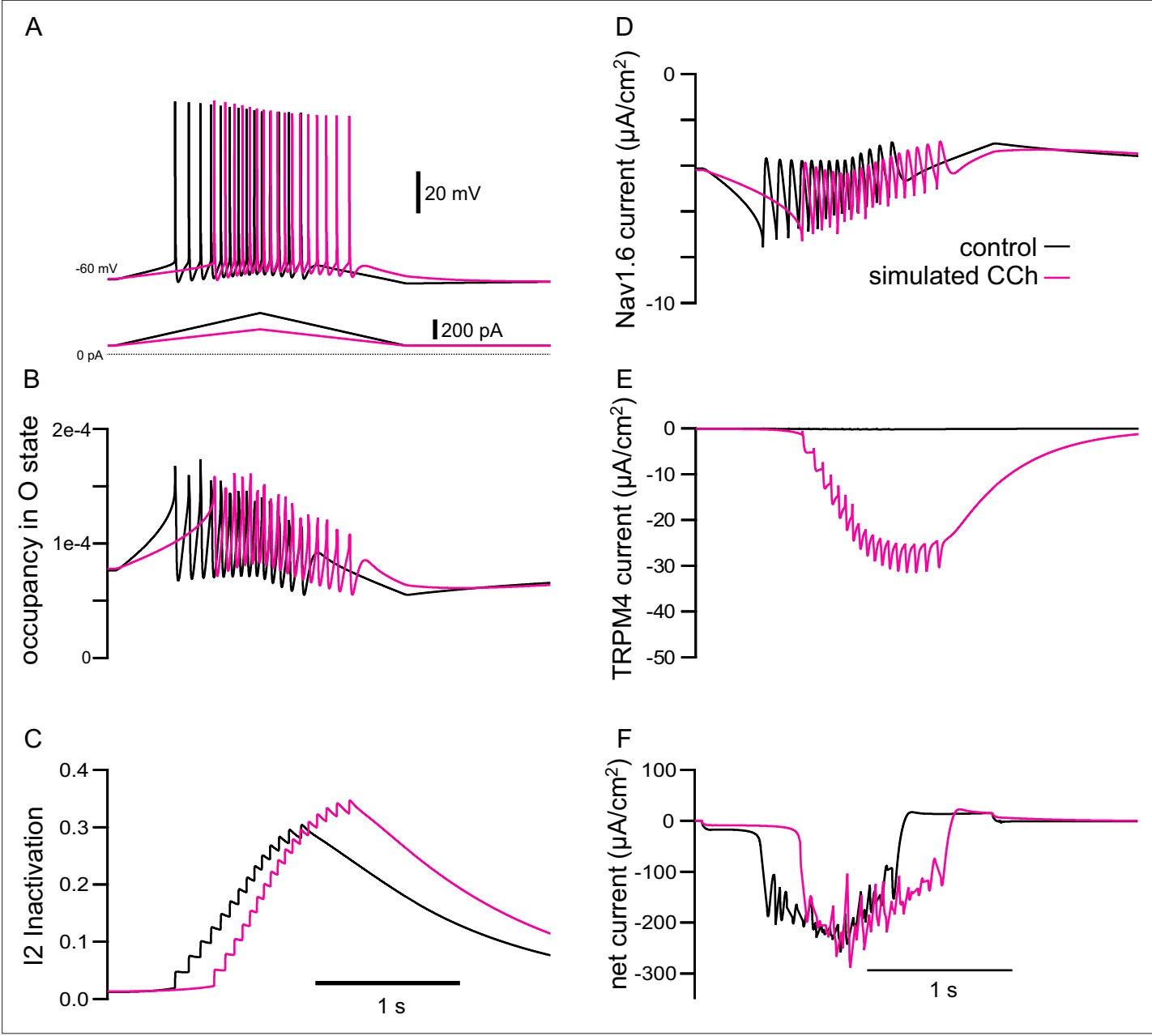

**Figure 8.** The model explains how simulated TRPM4 channel activation can reverse the adaptation mediated by long-term inactivation of Na$_V$1.6 channels. (**A**) Voltage traces from 7A1 and B1 overlaid; control (in black), CCh (in magenta). The triangular current injection ramps are shown below. (**B**) Occupancy of Na$_V$1.6 channel in the open state (O) in the somatic compartment, with values during spikes removed. (**C**) Occupancy of Na$_V$1.6 channel in the long-term inactivated state (I2). (**D**) Persistent Na$_V$ current (spiking values removed). (**E**) TRPM4 current. (**F**) Net ionic current. All values are shown for the somatic compartment.

## Discussion

### Summary of major results

Whereas place cell firing can only be observed in vivo, our bidirectional approach of modeling place cell firing in vitro and in silico can be precisely targeted to uncover molecular mechanisms responsible for modulation of firing patterns that lead to place cell firing in vivo. Manipulations that are difficult or impossible even in the in vitro preparation can be carried out in silico using our detailed multicompartmental model, with the added benefit that the model is transparent. All underlying state variables (ion

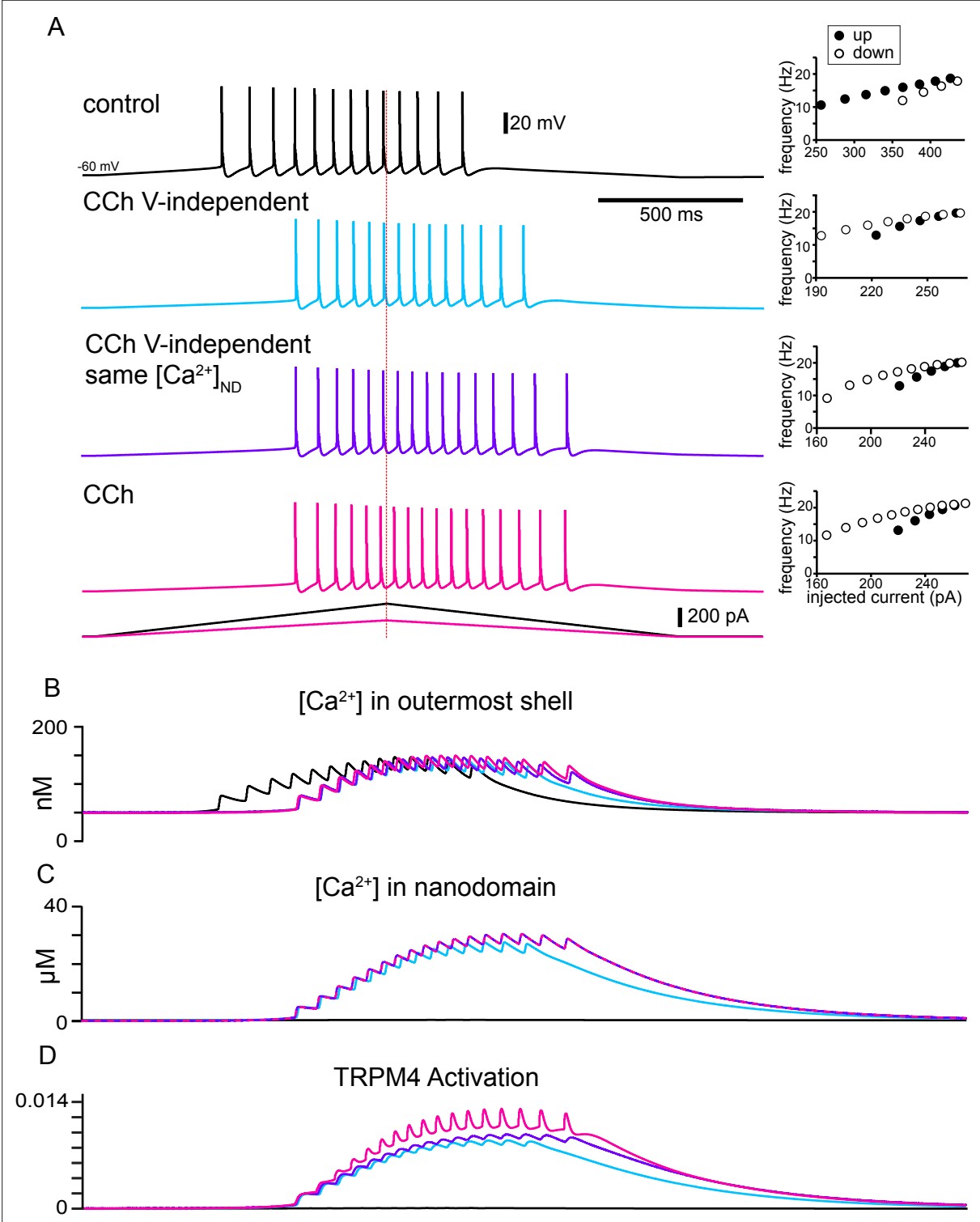

**Figure 9.** The model explains how activation of TRPM4 channels affects the firing pattern in response to triangular ramps that evoke similar firing frequencies. (**A**) Simulated voltage traces (top) during a two second triangular current ramp (just below voltage traces). Insets show the reciprocal of the interspike interval (ISI) plotted as the instantaneous frequency at the ISI midpoint. Vertical dashed red line shows the peak of the current ramp. The slower frequencies at similar values of injected current and fewer ISIs on the down-ramp (open circles) versus the up-ramp (closed circles) in control show hysteresis due to adaptation mediated by long-term inactivation of the sodium channel. TRPM4 channel activation causes the center of mass of firing to be shifted to the right in all other traces compared to control because a nanodomain for TRPM4 activation with special access to Ca²⁺ was added in simulated CCh. The hysteresis is in the opposite direction because the activation of TRPM4 channels by Ca²⁺ accumulation obscures and overwhelms

*Figure 9 continued on next page*

*Figure 9 continued*

adaptation. In the light blue traces, the voltage-dependence of TRPM4 activation was not included, so there is less $Ca^{2+}$ accumulation in the outer shell in (**B**) and in the nanodomain in (**C**), resulting in less activation of TRPM4 channels in (**D**). In order to isolate the effect of voltage-dependence from that of $Ca^{2+}$ dependence, the $Ca^{2+}$ waveforms in all nanodomains were recorded during the full CCh simulation shown in magenta, and played back into each nanodomain $Ca^{2+}$ during the simulations shown in purple.

concentrations, channel open fractions) are accessible so that mechanisms can be positively identified, facilitating design of experiments to test the putative mechanisms.

In this study we show that: (1) muscarinic receptor activation shifts the center of mass of firing from earlier to later during a depolarizing ramp, through (2) activation of $I_{CAN}$ mediated by TRP channels, most likely of the TRPM4 subtype; (3) the shift is not affected by the inhibition of either IP$_3$Rs or RyRs, arguing against a role of $Ca^{2+}$ released from the endoplasmic reticulum in this process; (4) multicompartmental modeling suggests a close physical relationship between the source of $Ca^{2+}$ increase and the TRPM4 channels; (5) the model prediction is supported by recordings showing that a very high concentration of BAPTA (30 mM), combined with buffering $[Ca^{2+}]_i$ at 10 nM are required to prevent the cholinergic-mediated rightward shift.

More broadly, our results suggest that some rapid changes to place cell firing could be triggered via neuromodulation of intrinsic excitability, in addition to and distinct from longer lasting experience-dependent changes that are brought about by synaptic and structural plasticity. Specifically, we speculate that increasing levels of ACh could lead to forward shifts in place cell firing, whereas decreasing levels of ACh could contribute to backwards shifts.

## Mystery current identified

Muscarinic activation of a $Ca^{2+}$-dependent nonselective cation current in rat CA1 pyramidal neurons has been known for nearly three decades (*Colino and Halliwell, 1993*; *Fraser and MacVicar, 1996*; *Dasari et al., 2017*). A previous study *Yamada-Hanff and Bean, 2013* found that in mouse CA1 neurons as well, muscarinic stimulation not only inhibits background potassium currents but also activates a nonselective cation current. They found that the latter effect was dominant and in fact converted CA1 pyramidal neurons into pacemakers with the application of 5–25 μM acetylcholine, 5–25 μM carbachol, or 5–10 μM oxotremorine-M. They suggested TRPC channels might mediate this current but were unable to positively identify the source of the nonselective cation current. Based on our results, this unidentified channel could be the TRPM4 channel. We show that TRPM4 channels primarily mediate a nonspecific $Ca^{2+}$-activated cation current evoked by cholinergic modulation in CA1 pyramidal neurons. Our model shows that the activation of TRPM4 channels contribute an additional inward current that activates and deactivates slowly upon depolarization. In the case of synaptically driven depolarizations (*Figure 2*), this inward current, with its slow kinetics (see *Figures 8 and 9*) is likely to cause an increased decay time of the EPSPs (see *Figure 2*). This aspect has likely implications for synaptic integration by these neurons, as shown by the observation of an impairment of LTP in TRPM4 knockout mice (*Menigoz et al., 2016*).

One of our most surprising findings was that TRPC channels did not seem to play a role in cholinergic modulation of the firing pattern along a triangular ramp. TRPC channels have been implicated in cholinergic signaling via direct activation by DAG, one of the two bioactive molecules generated by PLC hydrolysis of PIP$_2$ (the other being IP$_3$). Cholinergic-associated persistent firing, whereby neurons continue to fire after a depolarizing stimulus has ceased, has been attributed to activation of TRPC channels (*Zhang et al., 2011*; *Arboit et al., 2020*). However, the presence of cholinergic-induced persistent firing in TRPC knockout mice has called this into question (*Egorov et al., 2019*). In a recent study, TRPM4 channels were found to be involved in persistent firing generated upon $Ca^{2+}$ influx via T-type channels in thalamic neurons (*O'Malley et al., 2020*). Delayed bursting in a subtype of cerebellar granular cells was also shown to be mediated by TRPM4 channels (*Masoli et al., 2020*). Our study therefore provides further evidence regarding the role of these channels in the control of neuronal firing, possibly including persistent firing.

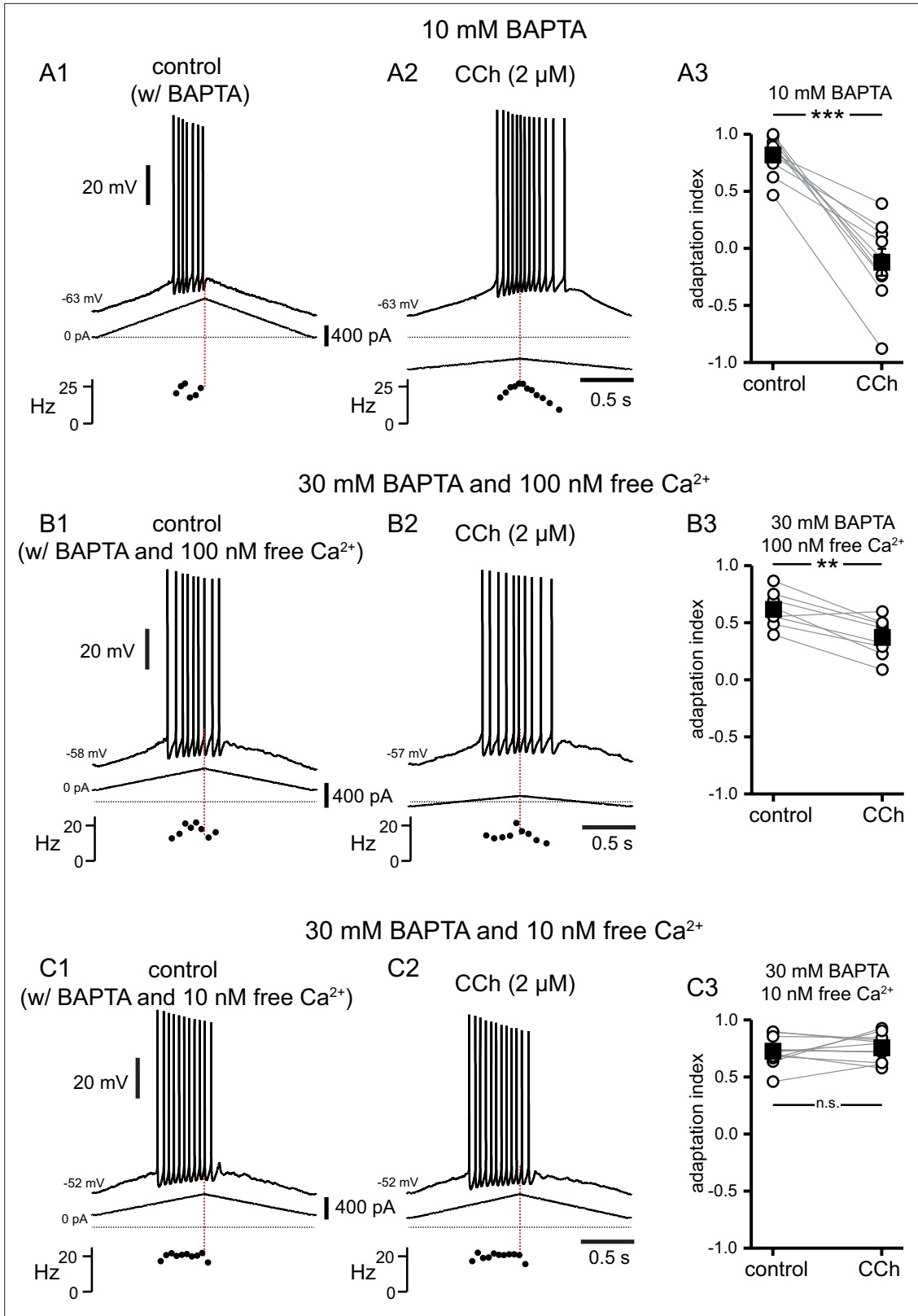

**Figure 10.** A high concentration of BAPTA is required to interfere with CCh-mediated shift in firing along ramp. (**A**) Somatic recordings during intracellular dialysis with 10 mM BAPTA, for a two second triangular current ramp injected in the soma before (**A1**), and during superfusion with CCh (2 µM) (**A2**). Current traces and instantaneous frequency plots are just below voltage traces. Red dotted line marks the middle of the ramp. Summary data (**A3**) of the adaptation index for control and CCh (n=10). Open circles connected by gray lines represent ratios for individual cells, before and

*Figure 10 continued on next page*

*Figure 10 continued*

during CCh. Black squares with error bars represent group averages ± SEM. (**B**) Same as A, but intracellular dialysis is with 30 mM BAPTA and 100 nM free [$Ca^{2+}$]. Summary data (**B3**) of the adaptation index for control and CCh (n=8). (**C**) Same as A, but intracellular dialysis is with 30 mM BAPTA and 10 nM free [$Ca^{2+}$]. Summary data (**C3**) of the adaptation index for control and CCh (n=12). *** $p<0.0005$, ** $p=0.001$, n.s. not significant, paired t-test. Source data in "*Figure 10—source data 1*".

The online version of this article includes the following source data for figure 10:

**Source data 1.** A high concentration of BAPTA is required to interfere with CCh-mediated shift in firing along ramp.

## Nanodomain resulting from colocalization of cholinergic-induced increases in [$Ca^{2+}$]$_i$ and TRPM4 channels

In HEK cells transfected with cDNA encoding TRPM4 channels, and in cardiac myocytes, activation of TRPM4 channels is secondary to IP$_3$ receptor activation (*Launay et al., 2002*; *Gonzales et al., 2010*; *Gonzales and Earley, 2012*), suggesting that TRPM4 channels in myocytes are activated by $Ca^{2+}$ release from intracellular stores. Furthermore, in situ proximity ligation assay was used to demonstrate co-localization of TRPM4 channels and IP$_3$Rs in human detrusor smooth muscle cells (*Provence et al., 2017*). In addition, in neurons of the preBötzinger nucleus TRPM4 channels are recruited by metabotropic glutamate receptors to generate robust inspiratory drive potentials through IP$_3$R activation (*Pace et al., 2007*). Surprisingly, we found that $Ca^{2+}$ release from the ER was not involved in the cholinergic shift of the center of mass of firing, since neither the IP$_3$R antagonist Xestospongin C (1–2 µM) nor the RyR antagonist ryanodine (40 µM) prevented the effects of carbachol. Therefore, although the mechanisms that produce the micromolar [$Ca^{2+}$] increases needed for TRPM4 channel activation are unknown, our results point to these channels acting as coincidence detectors, requiring both muscarinic receptor activation and depolarization-induced $Ca^{2+}$ influx during the ramp to be activated.

Our modeling results imply that, upon activation of muscarinic receptors, TRPM4 channels have privileged access to a nanodomain in order to achieve the micromolar concentrations of $Ca^{2+}$ necessary for their activation (*Nilius et al., 2004*), since microdomains (like the multicompartmental model's outer shell) only achieve submicromolar concentrations (*Fakler and Adelman, 2008*). The affinity of TRPM4 channels for $Ca^{2+}$ is heavily dependent on the availability of PIP$_2$ (*Nilius et al., 2006*) and binding of ATP and calmodulin or PKC-dependent phosphorylation (*Nilius et al., 2005*), which all function to increase the sensitivity of TRPM4 channels for $Ca^{2+}$. If under our conditions, the dependence of TRPM4 channels on [$Ca^{2+}$] has a much lower EC$_{50}$ and/or is much steeper than in the model we implemented (*Nilius et al., 2004*), perhaps the nanomolar [$Ca^{2+}$] in the outer shell could be sufficient to activate TRPM4 channels. However, the experimental result that 30 mM BAPTA was required to block the TRPM4 channels' access to $Ca^{2+}$ argues for a spatially restricted nanodomain. In a previous study, we found that intracellular BAPTA at a concentration 5 mM can mimic SK channel block (*Combe et al., 2018*), presumably by interfering with the calcium activation of those channels. The requirement of using 30 mM BAPTA while buffering free [$Ca^{2+}$]$_i$ to 10 nM in order to prevent TRPM4 channel activation suggests that TRPM4 channels are activated by a more spatially restricted $Ca^{2+}$ domain than SK channels.

## Convergence of pharmacological blockers

One caveat for our experimental results is that TRP channel blockers, like most drugs, have been reported to have some non-specific effects. In cardiac cells, both CBA and 9-phenanthrol have been shown to decrease the Ito1 potassium current, attributed to Kv4.3 (*Veress et al., 2018*; *Dienes et al., 2021*) and a late sodium current (*Hou et al., 2018*; *Dienes et al., 2021*). Kv4.3 is likely more important in interneurons than in pyramidal cells in the CA1 area (*Martina et al., 1998*), and neither of the two currents are affected by FFA at the concentration we used here (*Guinamard et al., 2013*). The other reported non-specific effects do not overlap between the three blockers, flufenamic acid, CBA and 9-phenanthrol (*Simard et al., 2012*; *Guinamard et al., 2013*; *Veress et al., 2018*; *Dienes et al., 2021*). The convergent effects of these three distinct drugs implicate TRPM4 channels as the most probable effector of the CCh-mediated modulation of the center of mass of firing. A similar pharmacological approach was used to implicate TRPM4 channel activation in subthreshold oscillations underlying pacemaker firing of chemosensory neurons in the retrotrapezoid nucleus (*Li et al., 2021*).

## Diverse cholinergic effects on CA1 pyramidal neurons

In CA1 pyramidal neurons, ACh binding to muscarinic receptors results in neuromodulation of neuronal excitability and presynaptic release probability. In our study, focused mainly on intrinsic properties, the predominant effect of ACh and CCh was to activate TRPM4 channels; however, the effect of ACh on CA1 pyramidal cells in vivo is multipronged. Muscarinic receptors are known to suppress several $K^+$ currents in CA1 pyramidal neurons (*Halliwell and Adams, 1982*; *Hoffman and Johnston, 1998*; *Buchanan et al., 2010*; *Giessel and Sabatini, 2010*). The effects of ACh on SK, M-type currents and $I_{CAN}$ are mediated by $M_1$ ($G_q$-coupled) receptors (*Dasari and Gulledge, 2011*). Synaptic depression of Schaffer collateral inputs to CA1 is mediated by $M_4$ ($G_i$-coupled) receptors (*Dasari and Gulledge, 2011*) located on presynaptic glutamatergic terminals, and synaptic depression of temporoammonic pathway glutamatergic inputs to CA1 is mediated by $M_3$ ($G_q$-coupled) receptors (*Palacios-Filardo et al., 2021*). When we used a synaptically driven depolarization (*Figure 2*), decreased release from Schaffer collateral was indicated by the smaller amplitude EPSPs in the presence of CCh. In these experiments, as in the current injection protocols, the center of mass of firing occurred earlier in the ramp in control and shifted to later in the ramp in the presence of CCh, suggesting that a combination of pre- and postsynaptic effects of cholinergic agonists can contribute to the shift. Moreover, the cholinergic down-regulation of these afferents justifies the use of smaller depolarizing current injection ramps in our recordings in the presence of carbachol, because less current injection is required to produce the same output frequencies due to enhanced intrinsic excitability in carbachol.

In a model of place field formation (*Savelli and Knierim, 2010*), if a neuron randomly fires within a place field (*Bittner et al., 2015*), its inputs from the currently active grid cells are strengthened and the input from those grid cells onto silent cells are weakened. Previously, ACh was thought to act in a manner consistent with volume transmission (*Dannenberg et al., 2017*). Recently, this view has been challenged by the finding that all hippocampal cholinergic terminals form synapses (*Takács et al., 2018*), and vesicles dock only at synapses. Since cholinergic neuromodulation of TRPM4 channels greatly increases excitability in CA1 pyramidal cells, selective synaptic ACh inputs may underlie the formation of place fields in a novel environment. Moreover, *Takács et al., 2018* found that the ACh terminals co-released GABA in separate vesicles, and that ACh acting on $M_2$ ($G_i$-coupled) autoreceptors on cholinergic terminals inhibited the release of both ACh and GABA. Future work is required to disentangle the distinct effects of ACh acting at the various muscarinic receptors ($M_1$-$M_5$).

## Acetylcholine, novelty and place field shifts

Acetylcholine acting at muscarinic receptors has been hypothesized to be a global novelty signal in the dentate gyrus (*Gómez-Ocádiz et al., 2022*), dorsal hippocampus (*Huff et al., 2022*), and the parahippocampal gyrus (*Frank and Kafkas, 2021*). A novel unconditioned stimulus caused a large increase in ACh extracellular levels in the hippocampus, but little locomotion (*Acquas et al., 1996*). On the other hand, hippocampal ACh release in an animal placed in a novel environment has two components, one related to novelty, that disappears in a familiar environment (*Thiel et al., 1998*), and one related to motor activity, associated with increased exploration (*Giovannini et al., 2001*). Indeed, a recent fiber photometric study showed that part of the population activity in medial septal cholinergic neurons is correlated with the logarithm of running speed (*Kopsick et al., 2022*). We are primarily interested in the component of the ACh signal related to novelty because of previous studies demonstrating modulation of place cell firing that was more pronounced during the first day in a novel environment (*Roth et al., 2012*).

In vivo recordings in rats demonstrated that place fields expand and the center of mass shifts backward with respect to the direction of travel on continuous tracks over several laps during each recording session (*Mehta et al., 2000*). This phenomenon was predicted by modeling studies that implemented Hebbian plasticity in the form of NMDA-mediated LTP (*Abbott and Blum, 1996*; *Blum and Abbott, 1996*), whereby spatially tuned input gradually strengthens over repeated traversals of a place field, causing a CA1 place cell to fire earlier in the place field. NMDA blockers were subsequently shown to occlude the backwards shift of the place field (*Ekstrom et al., 2001*) and an NMDA agonist partially restored it in aged rats (*Burke et al., 2008*). Calcium imaging in mice running on a treadmill in a virtual environment showed an abrupt backward shift in place cell firing (relative to where the first episode of place field firing occurred) during the first exposure to a novel environment (*Priestley et al., 2022*), as well as much smaller backward shifts in place fields in novel environments

that leveled off as the animals became more familiar with the environment during subsequent exposures (*Dong et al., 2021*). These backwards shifts, which were much larger than those observed previously and attributed to Hebbian plasticity, were ascribed to behavioral time scale plasticity, which is also dependent on NMDA receptor activation, and strengthens glutamatergic inputs active during an asymmetric seconds-long window skewed towards synapses activated before a burst triggered by a dendritic plateau (*Bittner et al., 2017*).

Despite the ability of elegant theories of synaptic plasticity to account for backward shifts in the center of mass of the place field, forward shifts have also been observed. Forward shifting of the center of mass of place field firing occurred, shifting toward prospective goal locations in the continuous T-maze alternation task as a reward was approached (*Lee et al., 2006*). Thus, task-specific demands may allow for the modulation of firing fields by neuromodulators such as ACh and dopamine. In a study that did not involve reward, but only exploration of a novel virtual linear maze (*Dong et al., 2021*), the center of mass of CA1 place fields in laps 2–4 were forward shifted compared to the first lap, for cells that already had a place field on the first lap (a lap required approximately 25 s). Laps 5 and beyond were backward shifted compared to the immediately preceding lap, so averaging over all runs, the place field was shifted backwards. Perhaps the neuromodulatory environment, including cholinergic tone, was different during the first four laps. The data in that study indicated that backward shifting in CA1 in a novel environment is due to a combination of place field expansion, skewness and translation of the point at which peak firing occurs. They suggested that since skewness and width dynamics are inconsistent across studies, multiple mechanisms may be involved in place field shifting.

## Ideas and speculations: Implications of our results for place fields in intact rodents

Our results suggest that increasing levels of ACh could lead to forward shifts in place field center of mass whereas decreasing levels of ACh could contribute to backwards shifts. Since cholinergic activity in CA1 is elevated when animals encounter novel environments, our results with CCh could correspond to an initial encounter with a novel environment, where firing of an existing place cell is shifted 'forward' or later during the depolarizing ramp that simulates spatially tuned input. Our control conditions, then, would be analogous to lower cholinergic tone, occurring as an animal becomes familiar with an environment, and the associated "backward" shift earlier in the depolarizing ramp. The rapid modulation of ion channels by removal of cholinergic tone could therefore provide a possible mechanism for the rapid shifting of place cell firing in a novel versus familiar environment, perhaps complementary to Hebbian plasticity that develops over time.

If our hypothesis is correct, optogenetic silencing of medial septum cholinergic afferents to CA1 during runs through a place field in a novel environment should partially occlude backwards shifts in place field center on subsequent runs that are due to diminished ACh in a familiar environment. Optogenetic stimulation of the same afferents during runs in a familiar environment should induce forward shifts in place field centers. Behaviorally, it could be informative to design experiments to test the effects of these manipulations on the animal's subjective sense of position, for example, demonstrated by overshoot or undershoot locations where the animal has been trained to pause.

There is a growing realization that there may be multiple mechanisms for place field shifting in addition to the well-known mechanisms for backwards shifts dependent on Hebbian plasticity (*Blum and Abbott, 1996*; *Ekstrom et al., 2001*; *Burke et al., 2008*; *Roth et al., 2012*; *Mehta et al., 2000*) or, more recently, behavioral timescale synaptic plasticity (*Bittner et al., 2017*; *Priestley et al., 2022*). We propose that neuromodulation of intrinsic properties by ACh, and potentially other neuromodulators like dopamine, provide an additional level of modulatory control in parallel with synaptic plasticity mechanisms, contributing to rapid place field shifts.

## Materials and methods

**Key resources table**

| Reagent type (species) or resource | Designation | Source or reference | Identifiers | Additional information |
|---|---|---|---|---|
| Strain, strain background | Male Sprague Dawley rats | Envigo (Inovit) | Hsd:Sprague Dawley SD | 7–12 weeks old when sacrificed for experiments |

*Continued on next page*

| Reagent type (species) or resource | Designation | Source or reference | Identifiers | Additional information |
|---|---|---|---|---|
| Chemical compound, drug | NBQX | Alomone Labs | cat# N-186 | AMPA receptor antagonist |
| Chemical compound, drug | DL-APV | HelloBio | cat# HB0252 | NMDA receptor antagonist |
| Chemical compound, drug | Gabazine | Alomone Labs | cat# G-216 | GABA$_A$ receptor antagonist |
| Chemical compound, drug | CGP55845 | Abcam | cat# ab120337 | GABA$_B$ receptor antagonist |
| Chemical compound, drug | Carbachol | Tocris | cat# 2810 | Acetylcholine receptor agonist |
| Chemical compound, drug | Acetylcholine | Sigma-Aldrich | cat# A6625 | Ligand for acetylcholine receptors |
| Chemical compound, drug | flufenamic acid | Sigma-Aldrich | cat# F9005 | TRP channel antagonist |
| Chemical compound, drug | SKF96365 | Alomone Labs | cat# S-175 | TRPC channel antagonist |
| Chemical compound, drug | CBA | Tocris | cat# 6724 | TRPM4 channel antagonist |
| Chemical compound, drug | 9-phenanthrol | Tocris | cat# 4999 | TRPM4 channel antagonist |
| Chemical compound, drug | Xestospongin C | Abcam | cat# ab120914 | IP$_3$R antagonist |
| Chemical compound, drug | low molecular weight heparin | Sigma-Aldrich | cat# H3149 | IP$_3$R antagonist |
| Chemical compound, drug | D-myo-IP$_3$ | Cayman Chemical | cat# 60960 | IP$_3$R agonist |
| Chemical compound, drug | Adenophostin A | MilliporeSigma | cat# 115500 | IP$_3$R agonist |
| Chemical compound, drug | Ryanodine | HelloBio | cat# HB1320 | RyR antagonist at [μM] |
| Chemical compound, drug | Ruthenium red | Tocris | cat# 1439 | RyR antagonist |
| Chemical compound, drug | Dantrolene | Tocris | cat# 0507 | RyR antagonist |
| Chemical compound, drug | caffeine | Tocris | cat# 2793 | RyR agonist |
| Chemical compound, drug | Thapsigargin | Alomone Labs | cat# T-650 | SERCA inhibitor |
| Chemical compound, drug | BAPTA tetra-potassium salt (K4-BAPTA) | Sigma-Aldrich | cat# A9801 | fast Ca$^{2+}$ chelator |
| Software, algorithm | MAXCHELATOR, WEBMAXC EXTENDED | https://somapp.ucdmc.ucdavis.edu/pharmacology/bers/maxchelator/webmaxc/webmaxcE.htm | | Online free metal calculator |
| Software, algorithm | SPSS | IBM | RRID:SCR_002865 | Statistics software |
| Software, algorithm | NEURON | *Hines and Carnevale, 1997*; *Carnevale and Hines, 2006* | doi:doi.org/10.1162/neco.1997.9.6.1179; doi.org/10.1017/CBO9780511541612 | Computational modelling and simulation software |
| Other | Laerd Statistics | https://statistics.laerd.com/ | | Online tutorial and software guide |

## Experimental methods

### Slice preparation

All the procedures described below were conducted according to protocols developed by following guidelines on the responsible use of laboratory animals in research from the National Institutes of Health and approved by the Louisiana State University Health Sciences Center-New Orleans Institutional Animal Care and Use Committee (IACUC, internal protocol numbers 3583 and 3851). Animals were housed in pairs, whenever possible; in the event of single housing, Bed-r'nest pucks (Andersons Lab Bedding) were provided for enrichment purposes.

For these experiments, 400-µm-thick slices were prepared from 7- to 11-week-old male Sprague Dawley rats as previously described (*Combe et al., 2018*). Briefly, rats were deeply anesthetized via intraperitoneal injection of ketamine and xylazine (90 and 10 mg/kg, respectively), until the disappearance of the toe-pinch and palpebral reflexes. After trans-cardiac perfusion with ice-cold solution, decapitation, and rapid removal of the brains, transverse hippocampal slices were cut using a vibratome. Slices were then transferred to a chamber filled with an oxygenated artificial cerebrospinal fluid (ACSF) containing, in mM: NaCl 125, $NaHCO_3$ 25, KCl 2.5, $NaH_2PO_4$ 1.25, $MgCl_2$ 1, $CaCl_2$ 2, dextrose 25, ascorbate 1, sodium pyruvate 3.

### Patch clamp electrophysiology

After recovery, individual slices were transferred to a submerged recording chamber, and superfused with ACSF at about 2 ml/min; the solution was warmed through an inline heater and measured to keep it at a temperature of 34–36°C in the chamber.

CA1 pyramidal cells were identified via differential interference contrast-infrared video microscopy. Whole-cell current clamp recordings were made using Dagan BVC 700 A amplifiers in the active 'bridge' mode. Recording pipettes were filled with an internal solution that contained, in mM: potassium methanesulphonate 125, KCl 20, HEPES 10, EGTA 0.5, NaCl 4, $Mg_2ATP$ 4, $Tris_2GTP$ 0.3, phosphocreatine 14. The electrode resistance was 1–3 MΩ for somatic recordings and 3–5 MΩ for dendritic recordings. Series resistance was monitored throughout the recordings and was usually less than 20 MΩ; recordings were discarded when series resistance reached 25 MΩ or 30 MΩ, for somatic and dendritic recordings, respectively. Cells with resting membrane potentials depolarized beyond –60 mV at break-in were discarded.

To investigate the contribution of intracellular $Ca^{2+}$ signaling to TRPM4 channel activation and the cholinergic shift in the center of mass of firing, in some sets of experiments, the internal patch clamp solution was modified as follows. In some experiments, EGTA was replaced by higher concentrations (10 or 30 mM) of the faster $Ca^{2+}$ chelator BAPTA tetra-potassium salt. When 30 mM BAPTA was included, 0.69 mM or 5.7 mM $Ca^{2+}$ was added to the internal solution to give 10 or 100 nM free $[Ca^{2+}]$, respectively, at 33 °C, pH 7.3 and an overall ionic concentration of 300 mM, according to the online program MAXCHELATOR, WEBMAXC EXTENDED version (https://somapp.ucdmc.ucdavis.edu/pharmacology/bers/maxchelator/webmaxc/webmaxcE.htm). In other experiments, Xestospongin C (1–2 µM) or ryanodine (40 µM) were added to the internal patch clamp solution to block $IP_3Rs$ and RyRs, respectively. Xestospongin C and ryanodine were dissolved in DMSO; in this case the [DMSO] in the final solution was ≤0.2% due to the limited solubility of these compounds. As in *Pace et al., 2007*, Xestospongin C was added to the standard intracellular solution immediately prior to use and discarded after 2 hr. In all these cases, intracellular dialysis appeared to take effect quickly, so the control data refer to the effect of the modified internal solution. Nonetheless, dialysis was allowed for at least 20 min in these experiments to make sure that the drug could diffuse into distal compartments.

NBQX (10 µM), DL-APV (50 µM), Gabazine (12.5 µM) and CGP55845 (1 µM) were applied in the external solution in all experiments, except those in which the Schaffer Collateral fibers were electrically stimulated (*Figure 2*), to block glutamatergic and GABAergic neurotransmission, respectively, in order to isolate the contribution of the intrinsic ion channels to the ramp responses. Carbachol (2 µM), acetylcholine (2, 10, or 15 µM), flufenamic acid (100 µM), SKF96365 (50 µM), CBA (50 µM), and 9-phenanthrol (100 µM) were added to the external solution as needed, from stock solutions made with water, DMSO, or ethanol; the concentration of ethanol or DMSO in the final solution was ≤0.1%. Carbachol, 9-phenanthrol, and CBA were purchased from Tocris, CGP55845 and Xestospongin C from Abcam (Cambridge, MA), DL-APV and ryanodine from HelloBio (Princeton, NJ), NBQX, Gabazine, and

SKF96365 from Alomone Labs (Jerusalem, Israel), and acetylcholine, flufenamic acid, and K4-BAPTA from Sigma (St. Louis, MO).

In order to approximate the depolarizing input that place cells receive as an animal traverses the place field, ramp-shaped depolarizing current injections were applied via the recording electrode to CA1 pyramidal neurons at either the soma or the apical dendrite (150–250 μm from soma). Two second ramps (1 s up, 1 s down) and ten second ramps (5 s up, 5 s down) were applied to simulate different running speeds. Current amplitude was adjusted to evoke peak frequencies between 10 and 25 Hz, comparable to place cell firing as recorded in vivo (*Hargreaves et al., 2007*; *Resnik et al., 2012*; *Bittner et al., 2015*).

Electrical stimulation was achieved by delivering constant current pulses through a tungsten bipolar electrode placed in the stratum radiatum of area CA1 to stimulate the Schaffer collateral fibers from CA3 pyramidal neurons. The instantaneous input frequency of stimulation was adjusted according to a linear, symmetric ramp (see also *Hsu et al., 2018*). The total duration of the ramp was ~2 s and the peak frequency at the center of the ramp was 25 Hz, our target for the maximum peak frequency in the ramp current injections (see above). The intensity of stimulation was adjusted such that neurons would fire in response to 40–65% of the total inputs and kept constant in control conditions and in the presence of carbachol. For each neuron and condition, we averaged the trials where the neurons did not fire in response to the first pulse in the synaptic ramp and the amplitude of the EPSPs was measured to estimate the degree of cholinergic modulation of presynaptic release. Antagonists of synaptic transmitter receptors were omitted in these experiments.

## Experimental design and statistical analyses

Experimental data were recorded and analyzed with Igor Pro software (WaveMetrics). Left or rightward shifts in firing along the current ramp were quantified as an adaptation index, as previously described (*Upchurch et al., 2022*):

$$\frac{\#APs \ on \ upramp - \#APs \ on \ downramp}{\#APs \ on \ upramp + \#APs \ on \ downramp}$$

Positive values indicate firing predominantly on the up-ramp, while negative values indicate firing mostly on the down-ramp. For synaptically driven membrane potential ramps, the adaptation index was calculated in a similar manner, where the up-ramp consisted of the first 15 (out of 30) stimuli. An adaptation index of zero indicates symmetry in the number of action potentials on the up-ramp and the down-ramp. Plots of instantaneous frequency versus time (f/t) or current (f/I) use the time or current value at the midpoint of the interspike interval (ISI), respectively. Current amplitude is relative to tonic baseline current. Raster plots in *Figures 1 and 2* show the timing of action potential firing for several trials in control conditions and in the presence of carbachol, where time = 0 indicates the beginning of the ramp. The tick marks represent the timing of each action potential (when the $V_m$ crossed –40 mV).

An n=6–12 is required for a typical patch clamp electrophysiology experiment that has a difference in means of 1.3–2 X and a standard deviation of 0.2–0.3 for p=0.05 and a power of 0.9 (*Cohen, 1977*). Each experimental group in this study is made up of n≥8 recording sessions; the number of cells recorded for each experiment is indicated in the Results section. Recordings were from one cell per slice; typically, a maximum of three recordings were obtained for each animal. Biological replicates (recording from different cells) were n≥8 for each experiment and (recordings from different animals) n≥6 for each experiment.

Statistical analyses were performed in SPSS (IBM, RRID:SCR_002865), following the tutorials and software guide from Laerd Statistics (2015, https://statistics.laerd.com/). Comparisons were made in the same cell before and during pharmacological treatment; the summary plots for these data include individual points and means ± standard error of the mean. Parametric analyses (paired samples t-test and repeated-measures ANOVA) were used to compare treatments in the same neurons, since data were normally distributed as determined by the Shapiro-Wilk test for normality. Significant differences demonstrated by repeated measures ANOVA were followed up by post-hoc analysis using Bonferroni-corrected pair-wise comparisons. Differences were considered to be statistically significant when p<0.05. Outliers, as assessed by boxplots, were not removed from analysis. To check the effect of the outliers, analyses were also performed after removing outliers, and with non-parametric tests

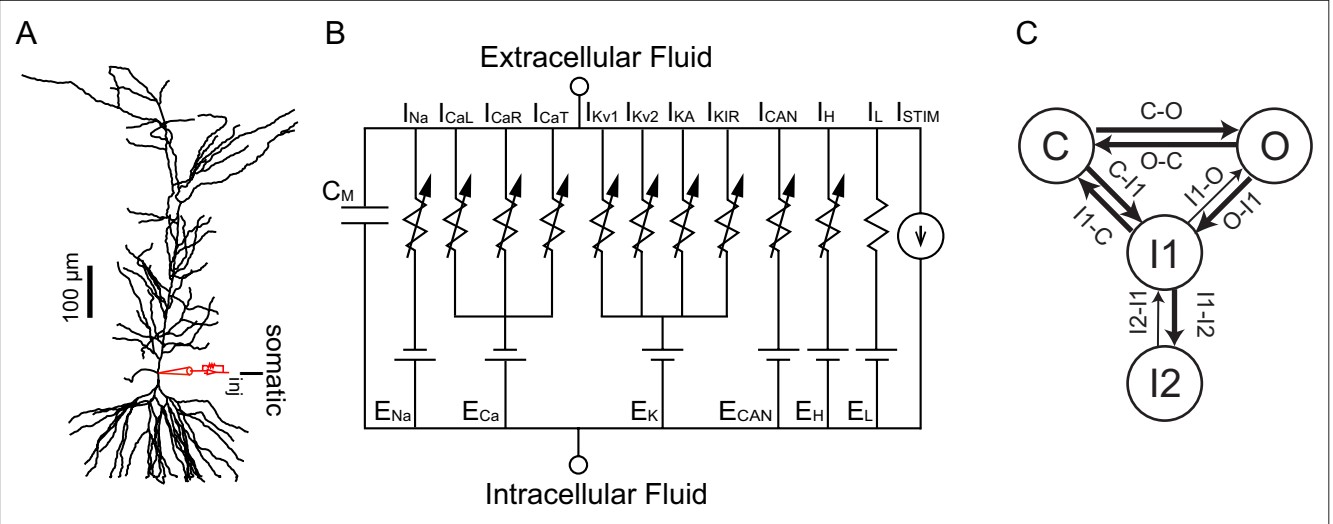

**Figure 11.** Model Schematics. (**A**) Reconstructed morphology ported to NEURON simulation program, with somatic current injection site labeled. (**B**) Equivalent circuit diagram, with the reversal potentials for each channel indicated by $E_X$. The following currents were modeled using Ohmic drive and conductances in parallel with a membrane capacitance ($C_M$): a Na$^+$ current ($I_{Na}$), L-type ($I_{CaL}$), R-type ($I_{CaR}$), and T-type ($I_{CaT}$) Ca$^{2+}$ currents, two delayed rectifiers ($I_{Kv1}$ and $I_{Kv2}$), an A-type K$^+$ current ($I_A$), an inward rectifying K$^+$ current ($I_{KIR}$), a nonspecific Ca$^{2+}$-activated current ($I_{CAN}$), a hyperpolarization-activated mixed cationic h-current ($I_h$), and a leak current ($I_L$). (**C**) Markov Model of the Na$_V$ channel with an open state (O), a closed state (C), a short-term inactivated state (I1) and a long-term inactivated state (I2). See also *Figure 11—figure supplement 1*.

The online version of this article includes the following figure supplement(s) for figure 11:

**Figure supplement 1.** Old model in which the source of Ca$^{2+}$ with privileged access to the nanodomain required to activated TRPM4 channels is IP$_3$R.

keeping the outliers. The significance did not change, and for consistency, the results of parametric tests are reported with all data included.

## Computational modeling

A multicompartmental CA1 pyramidal neuron model from our laboratory was used as a starting point (*Upchurch et al., 2022*). This model was based on the model in the study by *Poirazi et al., 2003a*, with subsequent changes made in studies by *Shah et al., 2008*, *Bianchi et al., 2012*, and *Combe et al., 2018*. The model (*Figure 11A*) was created from a reconstructed morphology (*Megías et al., 2001*) with 144 compartments and implemented in the NEURON simulation software package (*Hines and Carnevale, 1997*; *Carnevale and Hines, 2006*). The following currents (*Figure 11B*) were modeled using Ohmic drive and conductances in parallel with a membrane capacitance ($C_M$): a Na$^+$ current ($I_{Na}$), L-type ($I_{CaL}$), R-type ($I_{CaR}$), and T-type ($I_{CaT}$) Ca$^{2+}$ currents, two delayed rectifiers ($I_{Kv1}$ and $I_{Kv2}$), an A-type K$^+$ current ($I_A$), an inward rectifying potassium current ($I_{KIR}$), a nonspecific Ca$^{2+}$-activated current ($I_{CAN}$), a hyperpolarization-activated mixed cationic h-current ($I_h$), and a leak current ($I_L$). All currents were as in the previous model except ($I_{CAN}$), which was added to the model. In the soma and apical dendrites, we kept the Markov model (*Figure 11C*) of the Na$_V$ channel that we modified from *Balbi et al., 2017*. This Markov model was tuned to replicate the distance-dependent long-term inactivation of Na$_V$1.6 found in the literature (*Mickus et al., 1999*) that causes spike frequency adaptation and the difference in half activation between the soma and dendrites supported by *Gasparini and Magee, 2002*. The sevenfold gradient in the h-conductance supported by *Magee, 1998*, sixfold gradient in A-type potassium channels supported by *Hoffman et al., 1997* and lower sodium conductance in the dendrites versus the soma (75% of somatic value) supported by *Gasparini and Migliore, 2015* were also retained. A spine factor, to account for the additional membrane area due a higher density of spines (*Megías et al., 2001*) was varied as described in *Upchurch et al., 2022*, and a sigmoidal decrease in membrane and axial resistance was implemented to account for the decrease in input resistance along the apical dendrites (*Magee, 1998*; *Poirazi et al., 2003b*).

The calcium handling for the model (see top panels in *Figure 7A1 and B1*) was modified from *Ashhad and Narayanan, 2013*. The model performs a Ca$^{2+}$ material balance on four shells in each

compartment, and incorporates radial and longitudinal diffusion ($D_{Ca}\nabla^2\left[Ca^{2+}\right]$), stationary and mobile Ca$^{2+}$ buffers (R$_{Buf}$), calcium leak through endoplasmic reticulum leak channels (J$_{leak}$), Ca$^{2+}$ uptake from a sarcoplasmic/endoplasmic reticulum Ca$^{2+}$ ATPase pump (J$_{SERCA}$), IP$_3$-mediated Ca$^{2+}$ release (J$_{IP3R}$), a plasma membrane extrusion pump (J$_{Pump}$) and Ca$^{2+}$ from voltage gated channels on the membrane (J$_{VGCC}$). The latter two terms appear only in the outer shell. Details on the implementation of diffusion, the buffers, calcium leak and the pumps are described in *Ashhad and Narayanan, 2013*. The differential equation for cytosolic calcium is described below.

$$\frac{\partial\left[Ca^{2+}\right]}{\partial t} = D_{Ca}\nabla^2\left[Ca^{2+}\right] + AJ_{IP3R} + B\left(J_{leak} - J_{SERCA}\right) + R_{Buf} + J_{VGCC} - J_{Pump}$$

We used an equation for Ca$^{2+}$ dynamics in the nanodomain that is agnostic as to the source of the Ca$^{2+}$ contributing to the nanodomain:

$$\frac{d\left[Ca^{2+}\right]_{ND}}{dt} = X * I_{Ca} + \frac{\left[Ca^{2+}\right]_{outer\ shell} - \left[Ca^{2+}\right]_{ND}}{400ms}$$

where *X* is a scale factor representing the effect of a small volume for Ca$^{2+}$ accumulation in the restricted nanodomain. The model is agnostic regarding the source of the Ca$^{2+}$ entry into the nanodomain, which was arbitrarily set to be proportional to the total Ca$^{2+}$ current I$_{Ca}$. In the absence of simulated application of CCh, *X*=0 so that the TRPM4 channel senses the bulk Ca$^{2+}$ concentration in the outer shell. To simulate application of CCh, in contrast with our previous modeling efforts, based on a central role for IP$_3$R (see *Figure 11—figure supplement 1*), [IP$_3$] and therefore J$_{IP3R}$ was not increased from baseline levels; instead, influx to the nanodomain from an unknown source was simulated by setting *X*=500. The relatively long time constant reflected the spatially restricted nature of the nanodomain, with some postulated impediment to diffusion between the nanodomain and the bulk cytosol. In one simulation (purple traces in *Figure 9*), the [Ca$^{2+}$]$_{ND}$ for every segment in every section was recorded in the original CCh simulation (magenta traces) and played back using NEURON's vector play method in the model lacking the voltage-dependence of the TRPM4 channels in order to eliminate differences in [Ca$^{2+}$]$_{ND}$ between models and focus solely on the contribution of the voltage-dependence unrelated to additional Ca$^{2+}$ influx.

The TRPM4 model was modified from *Nilius et al., 2004* and assumes that the required binding of Ca$^{2+}$ to the channel precedes voltage-dependent channel activation. The model has three different states, an unbound state, a Ca$^{2+}$ bound closed state, and an open state.

$$TRPM4 \Leftrightarrow TRPM4\ bound\ to\ Ca^{2+} \underset{\alpha(V),\beta(V)}{\Leftrightarrow} Open\ TRPM4\ channel$$

Ca$^{2+}$ binding is assumed to be much faster than the voltage dependent gating and is assumed to instantaneously reach the steady state binding determined by the dissociation constant K$_d$ (87 μM). The fraction of open channels *m* relaxes to the steady state activation m$_\infty$ with a time constant, $\tau$, with first-order kinetics:

$$\alpha' = \frac{\alpha}{1 + \frac{Kd}{[Ca^{2+}]_{ND}}}, \quad m_\infty = \frac{\alpha'}{\alpha' + \beta} \quad \tau = \frac{1}{\alpha' + \beta}$$

where $\alpha$ is the forward rate of Ca$^{2+}$ bound TRPM4 channel opening, α(V)=0.0057 exp(0.0060 V), $\alpha'$ is the forward rate scaled by the fraction of Ca$^{2+}$ bound channels, β is the backward rate of TRPM4 channel closing, β(V)=0.033 exp(−0.019 V) and [Ca$^{2+}$]$_{ND}$ is the Ca$^{2+}$ concentration in the nanodomain. When removing the voltage-dependence of TRPM4 channels, we set V in the equations above to the steady-state value at –60 mV. We injected enough current into the model to hold the membrane potential at –60 mV long enough for all transients to equilibrate before injecting a two second temporally symmetric current ramp in the soma. The amplitude of the ramp was adjusted to reach similar peak frequencies as in the experiments, 10–25 Hz.

The persistent current during the interspike intervals determines the spike frequency. In order to focus on the rate determining currents in *Figures 8 and 9*, we removed points from the traces during

spiking by eliminating points at which the absolute value of the first temporal derivative of the current trace exceeded 0.005 µA/ms•cm$^2$ for the net current, 0.15 µA/ms•cm$^2$ for the TRPM4 current, and 0.2 µA/ms•cm$^2$ for the Na$_V$ current. Points outside the range of –0.3–0.09 µA/cm$^2$ in the net current trace, greater than 0 µA/cm$^2$ in the TRPM4 trace and less than –10 µA in the Na$_V$ current were also removed. Points were removed from the occupancy in O trace if the absolute value of the first temporal derivative of the O state exceeded 0.00002/ms or when occupancy exceeded 0.0002.

## Software accessibility

Model code is freely available and can be downloaded from ModelDB at: https://modeldb.science/267599.

## Acknowledgements

This work was funded by NIH R01 MH115832 under the CRCNS program to SG and CCC and by the Research Enhancement Program of the School of Medicine at the Louisiana State University Health Sciences Center to SG. The simulations utilized resources provided by NSF 2018936. We would like to thank Drs. Christopher Del Negro, Barbara Ehrlich and Dan Johnston for helpful discussions.

## Additional information

### Funding

| Funder | Grant reference number | Author |
| --- | --- | --- |
| National Institute of Mental Health | R01MH115832 | Carmen C Canavier<br>Sonia Gasparini |
| National Science Foundation | 2018936 | Carmen C Canavier |
| Louisiana State University Health Sciences Center School of Medicine | Research Enhancement Program | Sonia Gasparini |

The funders had no role in study design, data collection and interpretation, or the decision to submit the work for publication.

### Author contributions

Crescent L Combe, Data curation, Formal analysis, Investigation, Writing – original draft, Writing – review and editing; Carol M Upchurch, Software, Formal analysis, Investigation, Writing – review and editing; Carmen C Canavier, Conceptualization, Resources, Data curation, Software, Formal analysis, Supervision, Funding acquisition, Investigation, Methodology, Writing – original draft, Project administration, Writing – review and editing; Sonia Gasparini, Conceptualization, Resources, Data curation, Formal analysis, Supervision, Funding acquisition, Investigation, Methodology, Writing – original draft, Project administration, Writing – review and editing

### Author ORCIDs

Crescent L Combe ⓘ http://orcid.org/0000-0003-1181-6569
Sonia Gasparini ⓘ http://orcid.org/0000-0001-5847-9315

### Ethics

All the procedures described below were conducted according to protocols developed by following guidelines on the responsible use of laboratory animals in research from the National Institutes of Health and approved by the Louisiana State University Health Sciences Center-New Orleans Institutional Animal Care and Use Committee (IACUC, protocol numbers 3583 and 3851). Rats were deeply anesthetized via intraperitoneal injection of ketamine and xylazine (90 and 10 mg/kg, respectively), until the disappearance of the toe-pinch and palpebral reflexes.

Decision letter and Author response
Decision letter https://doi.org/10.7554/eLife.84387.sa1
Author response https://doi.org/10.7554/eLife.84387.sa2

## Additional files

### Supplementary files
• MDAR checklist

### Data availability

All data generated during this study are included in the manuscript; Source Data files have been provided for all experimental figures (*Figures 1–6* and *Figure 10*). Model code is freely available and can be downloaded from ModelDB at: https://modeldb.science/267599.

The following dataset was generated:

| Author(s) | Year | Dataset title | Dataset URL | Database and Identifier |
|---|---|---|---|---|
| Canavier CC, Upchurch CM | 2023 | Cholinergic Modulation Shifts the Response of CA1 Pyramidal Cells to Depolarizing Ramps via TRPM4 Channels with Potential Implications for Place Cell Firing (Combe et al., 2023) | https://modeldb.science/267599 | ModelDB, 267599 |

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
