## [Editor Report]

This manuscript by Combe et al. presents the role of cholinergic modulation in the spike rate adaptation in pyramidal place cells. Using combined electrophysiology, pharmacological, and multi-compartment computational modeling, the authors identify the downstream pathway (e.g. activation of TRPM4 channel) that shapes the firing pattern under the triangular-shaped ramps. The study demonstrates solid evidence, and these rigorous findings are important for bridging pyramidal neurons' molecular/channel properties to behavior-level implications (place field firing).

---

## [Decision Letter]

**Decision letter after peer review:**

Thank you for submitting your article "Cholinergic Modulation Shifts the Response of CA1 Pyramidal Cells to Depolarizing Ramps via TRPM4 Channels with Implications for Place Field Firing" for consideration by *eLife*. Your article has been reviewed by 3 peer reviewers, one of whom is a member of our Board of Reviewing Editors, and the evaluation has been overseen by John Huguenard as the Senior Editor. The following individual involved in the review of your submission has agreed to reveal their identity: J. Julius Zhu (Reviewer #3).

Essential revisions:

1) It has been shown recently that the asymmetric firing of CA1 place cells is due to synaptic weight changes resulting from synaptic plasticity (e.g., Bittner et al., 2017). This suggests that the asymmetric firing of place cells is primarily the result of asymmetric synaptic input. Therefore, the authors should test whether carbachol similarly affects a synaptically driven membrane potential ramp.

2) It has been shown before that the precision of spike timing depends on the stimulation pattern in vitro (Mainen and Sejnowski, 1995). The authors could add noise to their current stimulus and observe the effect on the AP firing patterns. If this is not possible, the authors should at least report the sweep-to-sweep variability for the data shown, e.g., in panels 1A2, 1B2, 1D2, and 1E2.

3) In most of the data presented in this manuscript, Carbachol appears to induce a 3 mV hyperpolarization and increase input resistance. As a result, the amount of current injected during Carbachol is drastically lower than during the controls. This should be emphasized more, and the input resistance should be quantified for each experimental condition. It should also be discussed whether this change in input resistance can account for the changes in the firing pattern observed. Finally, it should be clearly stated how the amount of the current injected was chosen for each cell, and data from a range of injected current ramps should be shown for each cell.

4) How the current result that TRPM4 channels can mediate the firing pattern change relates to the previous finding that the current injection evoked CA1 neuronal firing pattern is due to long-term Na channel inactivation (Upchurch, 2017, JNeurosci). What is the connection?

5) To validate the role of ER Ca^2+^ release in regulating TRPM, depletion of ER ca^2+^ pool with SERCA inhibitor (e.g. thapsigargin) would be a more direct way to test the model (also make sure to add TRPC inhibitor to avoid the store-operated Ca^2+^ entry).

*Reviewer #1 (Recommendations for the authors):*

1) Only panels 1A2, 1B2, 1D3, and 1E2 include x-axes that show the actual current values; in all the other figures, such axes are absent, which makes it hard to decipher the amount of current injected. Please change the plots in all the other figures to match the ones in figure 1.

*Reviewer #3 (Recommendations for the authors):*

To argue the cholinergic involvement in backwards shifts, the authors need first to show the cholinergic effect on down-ramp shifts. So far, the reported data (e.g., figure 1) imply that higher membrane potential, larger injected current, and longer duration (as shown in Epsztein et al. 2011 Neuron) all would cause larger increases in up-ramp firing, consistent with the authors' conclusion that activation of I_CAN_ is slow, and Ca^2+^ and voltage-dependent. If the authors believe otherwise, they should carry out more rigorous experiments using the physiological resting membrane potential, current strength, and duration and with the parameters properly controlled. In addition, they need to provide direct evidence to show increased acetylcholine release during backwards firing shifts and validate the cholinergic effect with optogenetic activation and inhibition. Alternatively, the authors might want to remove this claim.

---

## [Author Response]

Essential revisions:1) It has been shown recently that the asymmetric firing of CA1 place cells is due to synaptic weight changes resulting from synaptic plasticity (e.g., Bittner et al., 2017). This suggests that the asymmetric firing of place cells is primarily the result of asymmetric synaptic input. Therefore, the authors should test whether carbachol similarly affects a synaptically driven membrane potential ramp.

We thank the reviewers for this very important suggestion. We have added the results showing the effects of cholinergic modulation on a synaptically-driven membrane potential ramp, obtained by electrically stimulating the Schaffer collaterals with a stimulation frequency that was adjusted according to a linear, symmetric ramp (see also Hsu et al., Neuron 99,147-162, 2018). These results have been added to the manuscript in the Results section for new Figure 2 (lines 169-197) and in the Methods section (lines 716-726).

2) It has been shown before that the precision of spike timing depends on the stimulation pattern in vitro (Mainen and Sejnowski, 1995). The authors could add noise to their current stimulus and observe the effect on the AP firing patterns. If this is not possible, the authors should at least report the sweep-to-sweep variability for the data shown, e.g., in panels 1A2, 1B2, 1D2, and 1E2.

We addressed sweep-to-sweep variability among the various trials by adding raster plots of the timing of the spikes under control conditions and in the presence of CCh (Figure 1, new panels A3, B3, E3 and F3, see Results lines 128-134). We think that this is a very effective way of showing how the center of mass of firing shifts from early to late in the ramp in the presence of carbachol, and therefore we added the raster plots in the new Figure 2 as well, to report sweep-to-sweep variability under the inherently noisier conditions of synaptic stimulation.

3) In most of the data presented in this manuscript, Carbachol appears to induce a 3 mV hyperpolarization and increase input resistance. As a result, the amount of current injected during Carbachol is drastically lower than during the controls. This should be emphasized more, and the input resistance should be quantified for each experimental condition. It should also be discussed whether this change in input resistance can account for the changes in the firing pattern observed. Finally, it should be clearly stated how the amount of the current injected was chosen for each cell, and data from a range of injected current ramps should be shown for each cell.

We thank the reviewers for this comment, which made us realize that our initial presentation was not clear, in particular with regard to the traces that were chosen as examples in the initial submission of the paper. We now clarify on page 5 (lines 113-125) of the manuscript as follows:

“In some trials, under control conditions, we applied a baseline depolarization prior to the ramp, in order to capture the variability observed in vivo (Harvey et al. Nature 461:941–946, 2009; Epsztein et al. Neuron 70:109–120, 2011). Application of the cholinergic agonist carbachol (CCh, 2 µM) caused a depolarization of 2-6 mV. We compensated for this depolarization by injecting tonic hyperpolarizing current to reestablish the original membrane potential (see also Losonczy, et al., Nature 452, 436-442, 2008), as indicated by an offset from the 0 pA current level in the traces of the injected current ramps. The amplitude of background fluctuations in the resting membrane potential increased from a few tenths of a mV in control to 2-4 mV in CCh. Moreover, the threshold for action potential generation became more hyperpolarized. For all these reasons, we were not able to consistently vary the membrane potential using baseline depolarizations in the presence of CCh, because baseline depolarization alone frequently evoked spiking.”

For this reason, many of the carbachol example traces in the initial submission had more hyperpolarized V_m_ than their control counterparts. Acetylcholine also caused a depolarization in a dose-dependent manner, that was compensated for in the same way.

In this new version of the manuscript, we systematically report the effects of cholinergic agonists on membrane potential and neuronal excitability. Further, we show example traces with resting membrane potentials within 1 mV for each pharmacological comparison, therefore removing this variable and hopefully making results clearer. We also now state how the amount of injected current was chosen for each condition, and that the amount of injected current was generally lower in the presence of cholinergic agonists. Both the tonic hyperpolarizing current and the amplitude of the injected ramp for each example can now be appreciated in each figure.

Finally, the reviewers’ comment also made us realize that, in principle, the center of mass of firing could be systematically skewed by the initial membrane potential, the amplitude of the current ramp injection and/or the input resistance. For this reason, we added a supplementary figure (1-2) where the adaptation index was plotted as a function of each these variables. In all cases, it is apparent that the main factor determining whether the center of mass of firing is shifted earlier or later in the ramp is the presence or absence of carbachol rather than initial membrane potential, current injection amplitude, or input resistance.

4) How the current result that TRPM4 channels can mediate the firing pattern change relates to the previous finding that the current injection evoked CA1 neuronal firing pattern is due to long-term Na channel inactivation (Upchurch, 2017, JNeurosci). What is the connection?

We thank the reviewers for this suggestion, which helps to clarify our initial results. New Figure 8 addresses the connection between long-term inactivation of Na^+^ channels and the activation of TRPM4 channels, as characterized by the model (see Results lines 375-391). Furthermore, the model was instrumental in assessing how the ca^2+^ and voltage-dependence of TRPM4 channels synergize to contribute to the shift in the center of mass of firing (Figure 9). Figure 9 illustrates the positive feedback loop between ca^2+^ entry and the additional depolarization produced by ca^2+^ activation of TRPM4 channels that can potentially accelerate firing (see Results lines 392-427).

5) To validate the role of ER ca^2+^ release in regulating TRPM, depletion of ER ca^2+^ pool with SERCA inhibitor (e.g. thapsigargin) would be a more direct way to test the model (also make sure to add TRPC inhibitor to avoid the store-operated ca^2+^ entry).

For this resubmission, we have thoroughly investigated the role of the ER ca^2+^ pool, as suggested by the reviewers. Since reviewer 2 also suggested to pharmacologically dissect the role of IP_3_ vs ryanodine receptors, we have added a new figure to discuss these results. To our surprise, neither the IP_3_R antagonist, Xestospongin C (1-2 µM), nor the RyR antagonist ryanodine (40 µM) were effective in preventing the cholinergic shift of the center of mass of firing when added to the intracellular solution, as shown in new Figure 6 (see Results lines 311-340).

We also performed numerous pilot experiments using alternative approaches to assess the contribution of the ER ca^2+^ pool to the cholinergic-mediated activation of TRPM4 channels and the shift of the center of mass of firing to later in the ramp. In particular, we hypothesized that IP_3_R and/or RyR agonists could mimic the effect of CCh and that antagonists would prevent its effect. The pilot experiments are listed here:

IP_3_R agonistsD-myo-IP_3_ (50 µM, n = 6)adenophostin A (0.3-1 µM, n = 4)RyR agonistcaffeine (2-5 mM, n = 3)IP_3_R antagonistlow molecular weight heparin (1-3 mg/ml, n = 10)RyR antagonistsruthenium red (50-100 µM, n = 5)dantrolene (10-20 µM, n = 3)combination of IP_3_R antagonist and RyR antagonistXestospongin C (2 µM) + ruthenium red (100 µM, n = 2)Xestospongin C (2 µM) + ryanodine (100 µM, n = 2)depletion of ca^2+^ stores with SERCA inhibitorthapsigargin (10 µM, n = 4)

In all these cases, the pharmacological agents were added to the internal patch clamp solution. In the case of thapsigargin, the TRPC channel antagonist SKF96365 was added to the external solution, to avoid the store-operated Ca^2+^ entry.

The agonists did not mimic the cholinergic shift, and the antagonists and SERCA inhibitor did not prevent the cholinergic shift in the center of mass of firing. These experiments further confirm the data presented in Figure 6, but we chose to focus only on one pharmacological agent each for IP_3_Rs and RyRs for inclusion in the manuscript.

Reviewer #1 (Recommendations for the authors):1) Only panels 1A2, 1B2, 1D3, and 1E2 include x-axes that show the actual current values; in all the other figures, such axes are absent, which makes it hard to decipher the amount of current injected. Please change the plots in all the other figures to match the ones in figure 1.

Following this recommendation, we have modified all the experimental figures and we now show the actual current values injected for each cell in the figures, rather than the current amplitude relative to the holding current, as was presented in the initial submission. To this end, we have expanded the traces of injected ramp currents and plotted them as the absolute level of current injected, as referred to the 0 pA level. In this way, we think it is easier to estimate already at a glance the differences in the amount of current injected among the various treatments, as well as the tonic current that has to be injected in the presence of CCh or ACh to maintain the V_m_ at similar levels as in control conditions.

Reviewer #3 (Recommendations for the authors):To argue the cholinergic involvement in backwards shifts, the authors need first to show the cholinergic effect on down-ramp shifts. So far, the reported data (e.g., figure 1) imply that higher membrane potential, larger injected current, and longer duration (as shown in Epsztein et al. 2011 Neuron) all would cause larger increases in up-ramp firing, consistent with the authors' conclusion that activation of ICAN is slow, and ca^2+^ and voltage-dependent. If the authors believe otherwise, they should carry out more rigorous experiments using the physiological resting membrane potential, current strength, and duration and with the parameters properly controlled. In addition, they need to provide direct evidence to show increased acetylcholine release during backwards firing shifts and validate the cholinergic effect with optogenetic activation and inhibition. Alternatively, the authors might want to remove this claim.

This comment appears to be in line with some of the points raised above, in which reviewers expressed concern regarding the effect of CCh on membrane potential as well as the choice of the amplitude of injected current, and ramp length. We believe that the majority of these concerns were caused by our inadequate explanation and presentation of example traces in the first submission, and the lack of emphasis in the text on how the current injection amplitude was adjusted. Previous and new figures have been designed to clearly show V_m_, tonic current injection and ramp amplitude, and the text has been modified accordingly. As for ramp length, when determining the duration of the current injections for our ramps, we relied on the data recorded in vivo in freely moving rats (Epsztein et al. Neuron 70:109–120, 2011) or in head-fixed mice running on spherical a treadmill immersed in virtual reality (Harvey et al. Nature 461:941–946, 2009). In those papers, the voltage deflections are shown as a function of time, and gray bars or boxes represent the time the animals spend traversing the place field. We interpret those figures as showing that the hill-shaped depolarizations have variable durations, on the order of 1-20 s; we therefore think that our experiments with 2 and 10 second-long ramps cover a fair range of these durations. Our summary results in Figure 1 show that carbachol similarly affects 2 s and 10 s ramps.

We added a supplementary figure (1-2) where the adaptation index was plotted as a function of the initial membrane potential and ramp amplitude (as well as input resistance). In all cases, it is apparent that the main factor determining whether the center of mass of firing is shifted earlier or later in the ramp is the presence or absence of carbachol rather than initial membrane potential, current injection amplitude, or input resistance (see Results lines 158-168).

Further, we agree with the reviewer that our interpretation is somewhat speculative, and we have now included disclaimers as well as placed most of these interpretations in a portion of the discussion titled “Ideas and Speculations: Implications of our results for place fields in intact rodents”. In addition, we added the word “potential” in the title.